# Constrained Robust Submodular Partitioning

Shengjie Wang[*,1], Tianyi Zhou[*,1,2], Chandrashekhar Lavania[1] & Jeff A. Bilmes[1]

University of Washington, Seattle[1]; University of Maryland, College Park[2]
{wangsj,tianyizh,lavaniac,bilmes}@uw.edu

## Abstract

In the robust submodular partitioning problem, we aim to allocate a set of items into $m$ blocks, so that the evaluation of the minimum block according to a submodular function is maximized. Robust submodular partitioning promotes the diversity of every block in the partition. It has many applications in machine learning, e.g., partitioning data for distributed training so that the gradients computed on every block are consistent. We study an extension of the robust submodular partition problem with additional constraints (e.g., cardinality, multiple matroids, and/or knapsack) on every block. For example, when partitioning data for distributed training, we can add a constraint that the number of samples of each class is the same in each partition block, ensuring data balance. We present two classes of algorithms, i.e., Min-Block Greedy based algorithms (with an $\Omega(1/m)$ bound), and Round-Robin Greedy based algorithms (with a constant bound) and show that under various constraints, they still have good approximation guarantees. Interestingly, while normally the latter runs in only weakly polynomial time, we show that using the two together yields strongly polynomial running time while preserving the approximation guarantee. Lastly, we apply the algorithms on a real-world machine learning data partitioning problem showing good results.

## 1 Introduction

The problem of partitioning a given set $V$ of items into $m$ blocks, where any two blocks share no items in common, arises in many real-world scenarios and machine learning applications. As an optimization problem, partitioning aims to generate the blocks so that the utilities of the blocks, as measured by a given set function, are good. Submodular functions are a rich family of set functions that naturally captures diversity of a given set of items. They have been applied in many real-world problems [26, 30, 21, 17, 13, 18, 28]. By maximizing a submodular utility function for each partitioned block, we encourage each block to be representative of the ground set $V$. Many algorithms have been proposed for various submodular partitioning problems with approximation guarantees.

For the submodular welfare problem [24], we aim to find a partition such that the sum of the submodular evaluations of every block is maximized. Such an objective promotes the overall utility of the entire partition but some blocks may still have small function values. The robust submodular partitioning problem [12, 29] (often called "submodular fair allocation with indivisible goods") aims to find the partition such that the minimum-valued block in the partition is maximized according to the submodular function. The robust objective optimizes the worst block in the partition so that all blocks are minimally "good." In the general setting, every block in the partition may have a different submodular function (the heterogeneous case) although for this work, we study only the restricted setting where all blocks share the same submodular function (the homogeneous case). The robust submodular partitioning problem has many applications. Given $V$ as the training dataset for a machine learning task, Wei et al. [29] finds a partition of $V$ for distributed training: every block of partitioned data is sent to a single machine for parallel gradient computations, and the gradients

are aggregated over all the blocks in the partition for model updates. Since we enforce each block to be representative of $V$, the gradients computed across distributed machines are consistent, resulting in reduced variance and improved convergence for the aggregation step. Using a similar idea, Wang et al. [25] partitions the training data into mini-batches so that every mini-batch is as representative as possible, therefore reducing the variance during mini-batch gradient-based training.

In this work, we explore two different algorithmic approaches, *Min-Block Greedy* and *Round-Robin Greedy*, for our partitioning problem but under various constraints, newly applied to this problem. For Min-Block Greedy based algorithms, we first show that the $\frac{1}{m}$ bound for the unconstrained case is tight. We then modify the algorithm to allow a general down-closed constraint $\mathcal{C}$, and prove an approximation bound of $\frac{\alpha}{\alpha m+1}$, where $\alpha$ is the bound for solving the submodular maximization problem under constraint $\mathcal{C}$ using a greedy based algorithm. For example, for a cardinality constraint, $\alpha = 1-1/e$ [9], and the bound for constrained robust submodular partitioning is $(m + \frac{1}{1-1/e})^{-1}$. Similarly, for $\mathcal{C}$ as an intersection-of-$p$-matroids constraint, $\alpha = (1+p)^{-1}$ [10], and the bound is $(m+p+1)^{-1}$; for $\mathcal{C}$ as a knapsack constraint, $\alpha = 0.5(1-1/e)$ [16], and the bound is $(m+\frac{2}{1-1/e})^{-1}$. For Round-Robin Greedy based algorithms, when $\mathcal{C}$ is a cardinality constraint, we get a bound of $\frac{(1-1/e)^2}{3}$, and when $\mathcal{C}$ is a matroid constraint, we get a bound of $\frac{1-1/e}{5}$. The Min-Block Greedy approach gives a weaker bound, and since the $\frac{1}{m}$ bound for the unconstrained case is tight, we cannot improve upon the $\frac{1}{m}$ factor for the constrained case. The Round-Robin Greedy approach gives a constant bound, but its running time is worse. The running time for Min-Block Greedy is $\mathcal{O}(n^2)$, where $n$ is the ground set size. For Round-Robin Greedy under a matroid constraint, the running time is $\mathcal{O}(n^2(\log\log m + \log\frac{1}{\delta}))$, as it needs to use binary search to find the optimal solution value to the given problem over an exponentially decreasing sequence, with $\frac{1}{1+\delta}$ ($\delta > 0$) as the multiplicative factor. In all cases, we assume an oracle model, and the running time is in terms of the number of submodular evaluations. An important contribution our work shows is that by utilizing the Min-Block Greedy algorithm result as input, our Round-Robin Greedy algorithm attains **strongly polynomial running time** — all previous results on the unconstrained case using a Round-Robin-like algorithm have only weakly polynomial running time [3].

The various constraints (e.g., cardinality, matroids, and knapsack) we study greatly improves the applicability of robust submodular partitioning. Several applications that benefit from the constraints include: (1) Partition a training data for machine learning models in distributed training or forming deterministic mini-batches [29, 25]. The additional constraint can be the number of samples from each class to be no more than a certain value. If there are enough samples in the training data, every resulting block will have the same number of samples for each class, which avoids imbalance, further promoting each block's diversity, and improving the gradients' consistency. (2) Given an undirected graph, we partition the edges into subgraphs so that each subgraph is representative based on the submodular evaluation, and we also constrain each subgraph to have no cycles (a cycle matroid). A practical scenario is that we wish to send information efficiently over a graph of devices. We partition the graph so that information can be sent in parallel, and the constraint to have no cycles enforces that information is not redundantly sent twice to the same device, leading to improved communications efficiency. (3) Again, for an undirected and connected graph, we partition the edges into subgraph blocks such that if we were to remove any block of the partition from the original graph, the remaining graph remains connected (which can be done via a bond matroid, where min-cuts are cycles and anything not a cut is independent). In practice, this functions as a form of reliability insurance. For a graph of devices, we partition the graph to perform computation in parallel, so that if the connections in one block fail, the other blocks can still operate and communicate since the graph remains connected.

## 2 Related Work

Golovin [12] introduces robust submodular partitioning (i.e., submodular fair allocation of indivisible goods), and proposes a matching-based algorithm with a bound of $\frac{1}{n-m+1}$. Khot & Ponnuswami [14] proposes a binary search based algorithm and gives an improved bound of $\frac{1}{2m-1}$. Asadpour & Saberi [2] uses an ellipsoid approximation approach and gives a bound of $\Omega(\frac{1}{\sqrt{n}m^{1/4}\log n \log^{3/2} m})$. Wei et al. [29] gives a simple Min-Block Greedy algorithm and proves a $\frac{1}{m}$ bound. A Round-Robin Greedy approach is given in [3] with a bound of $\frac{1-e^{-1}}{3}$. Ghodsi et al. [11] proposes a local search algorithm with a bound of $\frac{1}{3}$. Both [3] and [11] requires guessing of the optimal solution value from an exponentially decreasing sequence of values, so strictly speaking, they lose an extra $(1+\delta)$-factor in the approxima-

tion bound where $(1 + \delta)$ is the exponential factor for the guessing sequence. We can set the $\delta$ value small to get close to the constant bounds shown above at the costs of computation. Wang et al. [25] extends the Min-Block Greedy algorithm with a cardinality constraint, and also shows a hierarchical partitioning framework to reduce the memory costs. We adapt the Min-Block Greedy approach [29] and the Round-Robin Greedy approach [3] to the constrained case. To the best of our knowledge, this work is the first (as far as we know) to study the robust submodular partitioning problem under all of the various constraints (cardinality, intersection of matroid, knapsack). Wang et al. [25] is a special case of our work as it only studies the cardinality constraint. Cotter et al. [8] studies (as well as allowing multiple blocks to be jointly scored) a matroid constrained "groupings" (i.e.,coverings, packings, or partitions) problem but only a fractional subset of groups (rather than the minimum of the groups), is guaranteed to have values larger than the bounded max-min OPT, while our bound compares the min block evaluation to the optimal max-min value. Another line of related research is the submodular load balancing problem, which minimizes the maximum-valued block in the partition according to the submodular evaluations. In contrast to promoting diversity of each block for the robust submodular partition problem, submodular load balancing encourages every block to contain redundant items, similar to standard clustering objectives. Theoretically, this problem has been shown to be much harder as Svitkina & Fleischer [22] shows a information theoretical lower bound of $o(\sqrt{\frac{n}{\log n}})$, and also gives a sampling algorithm to match the lower bound up to constant factors. Similar to the max-min case, Ghodsi et al. [11] uses the ellipsoidal approximation to get a bound of $\mathcal{O}(\sqrt{n} \log n)$. Wei et al. [29] gives a Lovász extension based relaxation algorithm and achieves a bound of $m$.

## 3 Preliminaries and Formulation

With a ground set $V$ of $n$ items, a submodular function $f$ is a set function $2^V \to \mathbb{R}$ that satisfies the property: $f(A) + f(B) \geq f(A \cup B) + f(A \cap B)$, where $A, B \subseteq V$. Equivalently, a submodular function is characterized by diminishing returns: $f(v|A) \geq f(v|B) \ \forall v \notin B$ and $A \subset B \subseteq V$, where $f(v|B) = f(\{v\} \cup B) - f(B)$. Submodular functions naturally describe the diversity or representativeness of a given set of items. Many simple greedy-based algorithms have been developed to solve optimization problems involving submodular functions, giving both theoretical approximation guarantees, as well as good empirical performance. We restrict the submodular functions discussed in this paper to be monotone non-decreasing and normalized, i.e., $f(B) \geq f(A) \ \forall A \subseteq B \subseteq V, f(\emptyset) = 0$.

A matroid $\mathcal{M} = (V, \mathcal{I})$ is a set system that describes the independence relationships among the subsets of the ground set $V$. $\mathcal{I}$ is a set of subsets of $V$ and every $S \in \mathcal{I}$ is considered an independent subset. The matroid rank function is defined as $r_{\mathcal{M}}(A) = \max\{|S| : S \subseteq A, S \in \mathcal{I}\}$. $r_{\mathcal{M}}(V)$ indicates the maximum size of a subset that may be independent according to the matroid $\mathcal{M}$. All subsets of cardinality $\leq k$ with some integer $k > 0$ form a *uniform matroid*, which we denote by $\mathcal{M}_k^u$. A partition matroid is one where we partition the ground set into blocks, and a set is independent if it intersects each block by no more than a block-specific limit. We define a particularly useful partition matroid on an expanded ground set $\bar{V}$ as follows: We first duplicate the ground set $m$ times, creating $V_1 = V_2 = \ldots V_m = V$, which are ground set copies. We create an expanded ground set $\bar{V} = \uplus_{j=1:m} V_j$ as the disjoint union. A subset $S \subseteq \bar{V}$ is independent in $\mathcal{M}_m^p$ if for every element $v \in V$, let its $m$ copies in $\bar{V}$ be $\{v_1, v_2, \ldots, v_m\}$, we have $|S \cap \{v_1, v_2, \ldots, v_m\}| \leq 1$, i.e., $S$ contains at most one copy of element $v$. Apart from the uniform matroid and this particular partition matroid, there are many other matroids reflecting a natural notion of independence, for example, the linearly-independent set of real vectors and the spanning trees in a graph. In the below, we use both $S \in \mathcal{M}$ and, when clear, $S \in \mathcal{I}$, to indicate that $S$ is independent in the matroid $\mathcal{M} = (V, \mathcal{I})$.

Matroids are often used as constraints in submodular optimization problems: $\max_{S \in \mathcal{I}} f(S)$ with a matroid $\mathcal{M} = (V, \mathcal{I})$. When $\mathcal{M}$ is a uniform matroid $\mathcal{M}_k^u$, this reduces to the cardinality submodular max and the greedy algorithm gives a $1 - e^{-1}$ bound [9]. For a general constraint with the intersection of $p$ matroids, the simple greedy algorithm gives a $\frac{1}{p+1}$ bound [10]. Suppose we represent a set $S$ as a binary indicator vector $x_S \in \{0, 1\}^n$, i.e., $\forall i \in [n], x_S[i] = 1$ if $v_i \in S$ or otherwise $x_S[i] = 0$. Then for all the independent sets of a matroid $\mathcal{M} = (V, \mathcal{I})$, the convex hull over all the $x_S, S \in \mathcal{I}$ forms a polytope, which is called the matroid polytope $\mathcal{P}_{\mathcal{M}}$ of matroid $\mathcal{M}$ [9]. Based on the convex property of the matroid polytope, algorithms [4–6, 24] have been proposed to firstly solve a continuous extension of the submodular optimization problem under the matroid polytope constraint, which generates a fractional solution in $[0, 1]^n$, and then round the fractional solution to an integral solution to get the resulting set. The continuous greedy algorithm [4] gives a $1 - e^{-1}$ guarantee under a single

matroid constraint using pipage rounding [1, 4]. Interestingly, running the continuous greedy under a partition matroid constraint (submodular welfare problem) gives a uniform fractional solution, i.e., on the expanded ground set $\bar{V}$, the fractional solution $x = (\frac{1}{m}, \frac{1}{m}, \ldots, \frac{1}{m})$ (i.e., assigning $\frac{1}{m}$ of every element to each block) leads to a $1 - e^{-1}$ bound in expectation for an integral solution that assigns each element in $V$ uniformly to one of the $m$ blocks. This observation also constitutes the basic idea of Round-Robin Greedy for solving the robust submodular partition problem [3], which we will discuss in more detail later.

For a submodular function $f$ on a ground set $V$, the robust submodular partition problem (submodular fair allocation) [12] is defined as:

$$\max_{\pi \in \Pi(V,m)} \min_{A \in \pi} f(A), \tag{1}$$

where $m$ is the number of blocks in a partition, we denote all possible partitions with $m$ blocks of ground set $V$ as $\Pi(V, m)$, and one partition $\pi$ with $|\pi| = m$ is a collection of $m$ disjoint sets. Equivalently, we can represent the partition using a partition matroid constraint on the expanded ground set $\bar{V}$:

$$\max_{S \subseteq \bar{V}, S \in \mathcal{M}_m^p} \min_{j \in [m]} f(S \cap V_j). \tag{2}$$

Intuitively, the above optimization for robust submodular partitioning encourages the minimum-valued block to have a high submodular evaluation. Compared to the submodular welfare problem, the robust submodular partition promotes fairness for every one of the partition blocks.

There have been three recent approximation algorithms developed to solve Eq. (1). Particularly, Wei et al. [29] uses a Min-Block Greedy algorithm, which greedily adds the element with the largest gain to the block with the minimum evaluation. Barman & Krishna Murthy [3] propose a Round-Robin Greedy algorithm, which iteratively traverses all the blocks in a fixed order, and greedily adds an element with the largest gain to each block. Ghodsi et al. [11] applies a local search approach, which starts with an arbitrary partition and keeps moving an element from a non-minimum block to the minimum block if this relocation improves the objective by certain threshold until no such element can be found.

For [3] and [11], they both require guessing the optimal solution's value, and they need to run multiple instances of their algorithms with the guessed optimal values as an exponentially decreasing sequence from the maximal possible value $f(V)$ to the optimal solution value $\mu = \max_{\pi \in \Pi(V,m)} \min_{S \in \pi} f(S)$. With the exponential decreasing factor as $1 + \delta$, the running time (in terms of submodular function calls) is $O(n^2 \frac{1}{\delta} \log \frac{f(V)}{\mu})$ for [3], and $O(n^2 m^2 \frac{1}{\delta} \log \frac{f(V)}{\mu})$ for [11]. Min-Block Greedy, a much simpler algorithm, has a running time of $O(n^2)$. Note that the settings of [3] and [11] are slightly more general than Eq. (1) as the submodular function for each block can be different. But it's not the heterogeneous case either as they focus on a different notion of optimality.

## 3.1 Discussions about the Optimality in [3, 11] and the Heterogeneous Case

The works [3, 11] both study the partitioning problem in the economics context of fair allocation of indivisible goods. In such a setting, every block is an agent, and we want to find an allocation of the goods to each agent in a fair manner, so that each agent's evaluation of the allocated goods to himself is optimized. Each agent can have different evaluations for the goods, which means that the submodular function for each block can be different. Taking [3] as an example, the theoretical guarantee they prove is

**Lemma 1** (Theoretical Guarantee in [3])**.** *Let $A_1, A_2, \ldots, A_m$ be the solution to the Round-Robin Greedy algorithm for the unconstrained problem, and given $m$ submodular functions for the $m$ blocks as $f_j$ for $j = 1, 2, \ldots m$, for every agent $j$ we have $f_j(A_j) \geq \frac{1 - e^{-1}}{3} \max_{\pi \in \Pi(V,m)} \min_{S \in \pi} f_j(S)$.*

Intuitively, the bound guarantees that based on each agent's evaluation $f_j$, the goods allocated to himself is not bad compared to the worst block in the allocation. When all the $f_j$'s are the same, the bound reduces to the bound for the homogeneous case of robust submodular partitioning, which is the focus of this paper. The heterogeneous case for robust submodular partitioning is different, as it requires to show a bound like (suppose the algorithm solution is $A_j$ for $j \in [m]$)

$$\min_{j \in [m]} f_j(A_j) \geq \gamma \max_{\pi \in \Pi(V,m)} \min_{S \in \pi} f_j(S). \tag{3}$$

We give an example of the $m$ functions so that the two guarantees vary. Say we have a predefined partition over the ground set $V$ into $m$ blocks as $C_1, C_2, \ldots, C_m$ and $\cup_{j \in [m]} C_j = V$, $C_j \cap C_{j'} = \emptyset$, $|C_j| = |C_{j'}|$ (assume $|V|$ is a multiple of $m$). Let $f_j(S) = |S \cap C_j|$. The optimal solution to the heterogeneous case $\max_{\pi \in \Pi(V,m)} \min_{j \in [m]} f_j(A_j)$ is to assign the items in the same way as the predefined partition $C_1, C_2, \ldots C_m$, and the optimal solution value is $|C_j|$. However, for the bound in [3, 11], the optimal solution $\{O_1, O_2, \ldots, O_m\} \in \mathrm{argmax}_{\pi \in \Pi(V,m)} \min_{A \in \pi} f_j(A)$ is to intersect each predefined block equally, i.e., $|O_j \cap C_{j'}| = \frac{|C_{j'}|}{m} \; \forall j, j' \in 1, \ldots, m$. Therefore, the optimal solution value is then $\frac{|C_j|}{m}$.

In this work, we study the constrained case for the submodular robust partition problem in the homogeneous setting (submodular functions for all blocks are the same). We extend Min-Block Greedy algorithm [29] and Round-Robin Greedy algorithm [3] to adapt to various constraints, e.g., cardinality, matroid, and intersection of matroids.

## 4 Min-Block Greedy Based Algorithms

Wei et al. [29] proposes a Min-Block Greedy Algorithm 2 for Eq. (1), which loops over $n$ iterations, and at every iteration, for the minimum-valued block $A_{j^*} \in \mathrm{argmin}_j f(A_j)$, it finds the element with the largest gain $f(v|A_{j^*})$. Wei et al. [29] proves a $1/m$ bound of Min-Block Greedy. In fact, their proof works for a simpler algorithm, Min-Block Streaming Algorithm 1, which assumes that the algorithm accesses elements from the ground set in an arbitrary order as a stream $V = (v_1, v_2, \ldots, v_n)$, and it assigns the incoming element to the block with the least evaluation. We denote the optimal partition to Eq. (1) as $\pi^* = \{O_1, O_2, \ldots, O_m\}$.

**Lemma 2** (**Unconstrained Min-Block Streaming[29]**). *For a ground set $V$ and its elements $(v_1, v_2, \ldots, v_n)$ coming in an arbitrary streaming order, the output solution of Alg. 1 has $\min_{j \in [m]} f(A_j) \geq \frac{1}{m} \min_{j \in [m]} f(O_j)$.*

**Corollary 1** (**Unconstrained Min-Block Greedy[29]**). *The output solution of Alg. 2 has $\min_{j \in [m]} f(A_j) \geq \frac{1}{m} \min_{j \in [m]} f(O_j)$ since the order of adding elements in Min-Block Greedy is one possible order of the ground set elements.*

Intuitively, Alg. 2 optimizes the objective Eq. (1) greedily, i.e., it always increases the current value (the minimum-block evaluation) with the largest possible gain, while the performance of Alg. 1 greatly depends on the order of elements, so it might seem that the bound for Min-Block Greedy should improve upon the current $\frac{1}{m}$ bound. However, as shown in our new result in the following lemma, the bound in Corollary 1 is tight.

**Lemma 3** (**Tightness of Corollary 1**). *$\forall \epsilon > 0, \exists$ a submodular function $f$ such that the output solution of Alg 2 $\min_{j=1:m} f(A_j) = \frac{1}{m} \min_{j=1:m} f(O_j) + \epsilon$.*

---

| **Algorithm 1:** Min-Block Streaming |
|---|
| **input** : submodular function $f$, ground set as a stream $V = (v_1, v_2, \ldots, v_n)$, number of blocks $m$ |
| 1 $R := V$; |
| 2 Let $A_1 = A_2 = \ldots = A_m = \emptyset$; |
| 3 **for** $i = 1 : n$ **do** |
| 4 $\quad j^* \in \mathrm{argmin}_j f(A_j)$; |
| 5 $\quad A_{j^*} := A_{j^*} \cup \{v_i\}$ ; |
| 6 **return** $(A_1, A_2, \ldots, A_m)$ |

| **Algorithm 2:** Min-Block Greedy |
|---|
| **input** : submodular function $f$, ground set $V$, number of blocks $m$ |
| 1 $R := V$; |
| 2 Let $A_1 = A_2 = \ldots = A_m = \emptyset$; |
| 3 **while** $R \neq \emptyset$ **do** |
| 4 $\quad j^* \in \mathrm{argmin}_j f(A_j)$; |
| 5 $\quad v^* \in \mathrm{argmax}_{v \in R} f(v|A_{j^*})$; |
| 6 $\quad A_{j^*} := A_{j^*} \cup \{v^*\}$ ; |
| 7 $\quad R := R \setminus \{v^*\}$; |
| 8 **return** $(A_1, A_2, \ldots, A_m)$ |

---

We elaborate on how to construct the submodular function in Appendix A. The key idea is that we can find a set-cover function where even though Min-Block Greedy selects the element with the largest gain, the element can still be quite redundant with the current minimum block. Say the current minimum block is $A$, and the maximum-gain element is $v$ chosen by the greedy step, meaning $f(v|A)$ is larger than $f(v'|A)$ for $v' \in R \setminus v$. However, $\frac{f(v|A)}{f(v)}$ can still be very small, i.e., the area covered by $v$ according to the set-cover function is already mostly covered by $A$. On the

other hand, the optimal solution can fully utilize $f(v)$ thus making a more lonesome $v$ cover a much larger area overall. Note that Lemma 3 also serves as the tightness for Lemma. 2 since the order of adding elements in Min-Block Greedy follows a streaming order.

More generally, given a constraint $\mathcal{C}$, we define the constrained robust submodular partition as:

$$\max_{\pi \in \Pi(V,m,\mathcal{C})} \min_{A \in \pi} f(A), \tag{4}$$

where $\Pi(V, m, \mathcal{C})$ is the set of all possible partitions on set $V$ into $m$ blocks such that for every partition $\pi \in \Pi(V, m, \mathcal{C})$, every block $A \in \pi$ should satisfy the constraint $A \in \mathcal{C}$. We denote the optimal partition in Eq. (4) as $\pi_C^* = \{O_1^{\mathcal{C}}, O_2^{\mathcal{C}}, \dots, O_m^{\mathcal{C}}\}$. We remark that due to the constraints, some elements might not be assigned to a partition block, so strictly speaking the solution is an allocation (or "grouping", Cotter et al. [8]) of elements rather than a partition.

For now, we will take $\mathcal{C}$ as any down-closed constraint: Let $\mathcal{C}$ be a collection of subsets of the ground set $V$, and by satisfying the constraint, we require the solution $A$ to be one of the subsets in $\mathcal{C}$. The down-closed property means that if $A \in \mathcal{C}$ we have $B \in \mathcal{C}$ for any $B \subseteq A$. Following Eq. (4), we can define the constrained problem in terms of the expanded subset $\bar{V}$:

$$\max_{S \subseteq \bar{V}, S \in \mathcal{M}_m^p, \forall j:(S \cap V_j) \in \mathcal{C}} \min_{j \in [m]} f(S \cap V_j). \tag{5}$$

Based on the Min-Block Greedy algorithm for the unconstrained case, we propose a natural extension to the constrained case (Alg. 3), where at every iteration, for the minimum-valued block $A_{j^*}$, we greedily find the best element $v^*$ that retains block feasibility under the constraint $\mathcal{C}$, i.e., $\{v^*\} \cup A_{j^*} \in \mathcal{C}$. If we cannot find any element in the remaining set to add to the current minimum block, we remove the current block from the candidate blocks and move to the next smallest valued block.

In Line 6 of Alg. 3, we call a subroutine GreedyStep$(R, \mathcal{C}, A_{j^*})$ to greedily find a feasible element. The subroutine varies according to the type of constraint $\mathcal{C}$. Particularly, for the constrained submodular maximization problem defined as

$$\max_{S \subseteq V, S \in \mathcal{C}} f(S). \tag{6}$$

GreedyStep$(\cdot)$ is shared by Alg. 3 and Alg. 4, and if Alg. 4 is an approximation algorithm of solving Eq. (6) with some bound $\alpha$, we can prove the following result for Alg. 3.

---

**Algorithm 3:** Constrained Min-Block Greedy

**input** : submodular function $f$, ground set $V$, number of blocks $m$, constraint $\mathcal{C}$

1 Let $A_1 = A_2 = ... = A_m = \emptyset$;
2 Let $J = [m], R = V$;
3 **while** $R \neq \emptyset$ *and* $J \neq \emptyset$ **do**
4     $j^* \in \arg\min_{j \in J} f(A_j)$;
5     **if** $\exists v \in R$ *s.t.* $A_{j^*} \cup \{v\} \in \mathcal{C}$ **then**
6        $v^* := $ GreedyStep$(R, \mathcal{C}, A_{j^*})$;
7        $A_{j^*} := A_{j^*} \cup \{v^*\}, R := R \setminus \{v^*\}$;
8     **else**
9        $a^* \in \arg\max_{a \in A_{j^*} \cup R} f(\{a\})$;
10        **if** $f(\{a^*\}) \geq f(A_{j^*})$ **then**
11           $A_{j^*} := \{a^*\}, R := R \setminus \{a^*\}$ ;
12        Let $J = J \setminus j^*$;
13 **return** $(A_1, A_2, ..., A_m)$

---

**Algorithm 4:** Constrained Submodular Greedy Max

**input** : submodular function $f$, ground set $V$, constraint $\mathcal{C}$

1 $R := V$;
2 Let $S^g = \emptyset$;
3 **while** $R \neq \emptyset$ **do**
4     **if** $\exists v \in R$ *s.t.* $A_{j^*} \cup \{v\} \in \mathcal{C}$ **then**
5        $v^* := $ GreedyStep$(R, \mathcal{C}, S^g)$ ;
6        $S^g := S^g \cup \{v^*\}$ ;
7        $R := R \setminus \{v^*\}$;
8     **else**
9        Break;
10 $a^* \in \arg\max_{a \in V} f(\{a\})$;
11 **return** $\arg\max_{A \in \{S^g, \{a^*\}\}} f(A)$

---

**Theorem 1 (Constrained Min-block Greedy).** *Given a constraint $\mathcal{C}$, if the greedy solution $S^g$ to problem $\max_{S \in \mathcal{C}} f(S)$ using Alg. 4 has a bound of $\alpha$, i.e., $f(S^g) \geq \alpha \max_{S \in \mathcal{C}} f(S)$, then the solution of Alg. 3 has $\min_{j \in [m]} f(A_j) \geq \frac{\alpha}{\alpha m + 1} \min_{j \in [m]} f(O_j^{\mathcal{C}})$. Assuming GreedyStep$(\cdot)$ takes $\mathcal{O}(1)$ oracle calls, the time complexity of of Alg. 4 is $\mathcal{O}(n^2)$.*

The general idea of the proof (details in Appendix A) is that we divide the ground set $V$ into two disjoint parts $V = V' \cup R'$, where the min block in the output solution $A$ intersecting $V'$ corresponds

to the min block solution of an instance of unconstrained robust partition problem (Eq. (1)) defined on the ground set $V'$, and $A \cap R'$ with the other part corresponds to the solution of an instance of submodular maximization (Eq. (6)) under the constraint $\mathcal{C}$ defined on the ground set $A \cup R'$. We bound the two parts separately and combine them to obtain the above bound.

**Corollary 2** (**Cardinality Constrained Min-block Greedy**). *For $\mathcal{C}$ as a cardinality constraint, the output of Alg 3 has* $\min_{j=1:m} f(A_j) \geq \frac{1}{m + \frac{1}{1-e^{-1}}} \min_{j=1:m} f(O_j^{\mathcal{C}})$.

**Corollary 3** (**Matroid Constrained Min-block Greedy**). *For $\mathcal{C}$ as an intersection of $p$ matroids constraint, the output of Alg 3 has* $\min_{j=1:m} f(A_j) \geq \frac{1}{m+p+1} \min_{j=1:m} f(O_j^{\mathcal{C}})$.

**Corollary 4** (**Knapsack Constrained Min-block Greedy**). *For $\mathcal{C}$ as a knapsack constraint, the output of Alg 3 has* $\min_{j=1:m} f(A_j) \geq \frac{1}{m + \frac{2}{1-1/e}} \min_{j=1:m} f(O_j^{\mathcal{C}})$.

For $\mathcal{C}$ as a cardinality constraint, GreedyStep($\cdot$) just picks the element with the largest gain assuming the block has not yet reached the cardinality limit $k$. For $\mathcal{C}$ as an intersection of $p$ matroid constraints, GreedyStep($\cdot$) finds the element $v^*$ that has the largest gain $f(v^*|A_{j^*})$ assuming the block can be kept feasible, i.e., $v^* \cup A_{j^*} \in \mathcal{C}$. For $\mathcal{C}$ as a knapsack constraint with the weight of each element $v$ as $w(v)$, GreedyStep($\cdot$) finds the element $v^*$ with the largest ratio $\frac{f(v^*|A_{j^*})}{w(v)}$ assuming the sum of weights can be kept below the given budget. In line 9-11 of Alg. 3 and line 10-11 of Alg. 4, we include an extra step of comparing with the largest singleton value. Such step is redundant when $\mathcal{C}$ is an intersection of matroid constraints, but is essential for the knapsack constraint case, as the modified greedy algorithm for the knapsack problem [16] requires this extra step or otherwise $\alpha$ is unbounded. Due to the tightness of the $\frac{1}{m}$ bound we have proved for the unconstrained case, the $\frac{1}{m}$ dependence in the constrained bound cannot be improved.

## 5 Round-Robin Greedy Based Algorithms

Barman & Krishna Murthy [3] propose a round-robin style algorithm for the unconstrained robust submodular partition problem (Eq. (1)) and gives a constant bound of $\frac{1-e^{-1}}{3}$ with weakly polynomial running time. Compared to Min-Block Greedy, Round-Robin Greedy requires guessing the optimal values by an exponentially decreasing sequence, and for each guessed value, it runs one instance of the round-robin subroutine. Specifically, suppose $\mu = \min_{j \in [m]} f(O_j)$, i.e., $\mu$ is the optimal solution value for the unconstrained case, then for a parameter $\delta > 0$, Round-Robin Greedy runs the round-robin subroutine with the guessed optimal values from a sequence $(f(V), \frac{f(V)}{1+\delta}, \frac{f(V)}{(1+\delta)^2}, \ldots)$ and ends when the guessed value is no larger than $\mu$. The running time of each round-robin subroutine is $\mathcal{O}(n^2)$, as it greedily finds the element with the largest gain by iterating over all the remaining elements. There are $\log_{1+\delta} \frac{f(V)}{\mu}$ guessed values in the exponentially decreasing sequence, so the overall running time is $\mathcal{O}(n^2 \log_{1+\delta} \frac{f(V)}{\mu}) = \mathcal{O}(n^2 \frac{1}{\delta} \log \frac{f(V)}{\mu})$. Note that since we use a $(1+\delta)$ factor exponentially decreasing sequence, we lose a $(1+\delta)$ factor in the approximation bound, which can be improved arbitrarily by using a smaller $\delta$ value but with a cost of running more instances of the round-robin subroutine.

The major idea behind Round-Robin Greedy comes from the solution of Continuous Greedy for the submodular welfare problem, which is an uniform fractional vector $x = (\frac{1}{m}, \frac{1}{m}, \ldots, \frac{1}{m})$ with the length of $x$ equal to the size of the expanded ground set $|\bar{V}|$. Let $F$ be the multilinear extension [4] of $f$, Continuous Greedy gives a bound that $F(x) = \mathbb{E}_{R \sim x} f(R) \geq (1 - e^{-1}) \max_{\pi \in \Pi(V,m)} \sum_{A \in \pi} f(A)$, where $\mathbb{E}_{R \sim x} f(R)$ takes the expectation of $f(R)$ on a random set $R$ with each element sampled independently according to the probability in the fractional vector $x$. Note that the hardness for submodular optimization under a matroid constraint is $1 - e^{-1}$, which means that the random assignment strategy achieves the best possible theoretical bound on the submodular welfare problem.

Round-Robin Greedy can be thought as a rounding mechanism for the fractional solution $x$. The round-robin style iteration is similar to the uniform random assignment in a deterministic manner, and by greedily finding the element, the value of every block can be bounded against that for the random assignment. In fact, Round-Robin Greedy bounds every block $A_j$ to be $f(A_j) \geq \frac{1}{3} \frac{F(x)}{m}$, and since the welfare solution bounds the robust solution in terms of the sum: $\max_{\pi \in \Pi(V,m)} \sum_{A \in \pi} f(A) \geq \sum_{j \in [m]} O_j \geq m\mu$, we get the desired bound for the robust partition problem.

We extend Round-Robin Greedy to the constrained case (Eq. (4)) firstly with $\mathcal{C}$ as a cardinality constraint $k$. This is a relatively simple case due to the nature of Round-Robin Greedy that every block gets assigned with the same number of elements at the end of every round-robin iteration. We present the modified algorithm in Alg. 5, which also helps to explain the essential ideas of the original Round-Robin Greedy as we describe below.

**Lemma 4 (Cardinality Constrained Round-Robin).** *For the problem in Eq. (4), with $\mathcal{C}$ as a cardinality constraint $k$, Alg. 5 gives a solution $\min_{j\in[m]} f(A_j) \geq \frac{(1-e^{-1})^2}{3} \min_{j\in[m]} f(O_j^k)$.*

---

**Algorithm 5:** Cardinality Round-Robin Greedy

**input** : $f$, $V$, $m$, cardinality constraint $k$, discounting factor for guessing optimal $\delta$

1   Let $\tau$ be the solution value of Alg. 3;
2   Let $high = \lceil \log_{1+\delta}(m+2) \rceil$, $low = 0$;
3   Create a sequence of guessed values: $(\tau, (1+\delta)\tau, (1+\delta)^2\tau, \ldots, (1+\delta)^{high}\tau)$;
4   Create an empty solution ($\emptyset$ for each block in the partition) for each guessed value
    $\pi_0, \pi_1, \ldots, \pi_{high}$;
5   **while** $high \geq low$ **do**
6     Let $idx = \lfloor (high+low)/2 \rfloor$; Let $A_1 = A_2 = \ldots = A_m = \emptyset$;
7     Let $V' = \{v | v \in V, f(v) \leq \frac{(1-e^{-1})^2}{3}(1+\delta)^{idx}\tau\}$; Let $G = V \setminus V'$;
8     Assign $G$ to $A_{m-|G|+1}, A_{m-|G|+2}, \ldots, A_m$ with one element per block;
9     Let $m' = m - |G|$;
10     Let $A_1', A_2', \ldots, A_{m'}'$ be the solution to $\max_{\pi\in\Pi(V',m',k)} \sum_{S\in\pi} f(S)$ using continuous greedy and swap rounding; Let $V'' = \cup_{j\in[m]'} A_j'$;
11     Let $\{A_1, A_2, \ldots, A_{m'}\} = RR(f, V'', m', \mathcal{M}_k^u, [m'])$;
12     **if** $f(A_j) \geq \frac{(1-e^{-1})^2}{3}(1+\delta)^{idx}\tau \; \forall j \in [m']$ **then**
13       Let $\pi_{idx} = \{A_1, A_2, \ldots, A_m\}$; Let $low = idx + 1$;
14     **else**
15       Let $high = idx - 1$;
16   **return** best of $\pi_0, \pi_1, \ldots, \pi_{high}$;

---

Here is how we achieve strongly-polynomial time. Different from the original Round-Robin Greedy, which performs a grid search over guessed optimal values, we perform a binary search over the sequence of values and therefore the number of outer iterations is reduced. Most importantly, we use the Min-Block Greedy solution's value as the minimum guessed value $\tau$. Because of the $\frac{1}{m+\frac{1}{1-1/e}}$ bound of the Min-Block Greedy solution, the maximum guessed value is thus bounded by $(m+2)\tau$. We then create a $1+\delta$-factor exponential decreasing sequence between $\tau$ and $(m+2)\tau$ to binary search for the optimal solution value. This improves the number of outer iterations of the algorithm to $\mathcal{O}(\log\log_{1+\delta} m) = \mathcal{O}(\log\log m + \log\frac{1}{\delta})$, which is strongly-polynomial while the number of outer iterations $\mathcal{O}(\log_{1+\delta}\frac{f(V)}{\mu})$ for the original unconstrained case is only weakly-polynomial as it has a log dependence on the function value.

**Algorithm 6:** Round-Robin Greedy Iterations $(RR(f, R, m', \mathcal{M}, J))$

**input** : $f$, $R$, $m'$, matroid constraint $\mathcal{M}$,
       set of block indices $J$

1   **while** $J \neq \emptyset$ and $R \neq \emptyset$ **do**
2     **for** $j \in [m']$ **do**
3       **if** $j \in J$ **then**
4         **if** $\exists v \in R$ s.t. $A_j \cup \{v\} \in \mathcal{M}$
         **then**
5           $v^* \in$
           $\underset{v\in R, A_j\cup v\in\mathcal{M}}{\arg\max} f(v|A_j)$;
6           $A_j := A_j \cup \{v^*\}$ ;
7           $R := R \setminus \{v^*\}$;
8         **else**
9           Let $J = J \setminus j$;
10   **return** $(A_1, A_2, \ldots, A_{m'})$

In every outer iteration (Line 12-15), Alg 5 checks if the round-robin solution based on the guessed optimal value $(1+\delta)^{idx}\tau$ satisfy the approximation bound, i.e., $f(A_j) \geq \frac{(1-1/e)^2}{3}(1+\delta)^{idx}\tau$ $\forall j \in [m]'$. If the bound is (not) satisfied, the guessed value is large (small) and we move to an increased (decreased) search value. Within every outer iteration, we perform round-robin greedy (iterate over every block in some fixed order and greedily add to the block the element with the largest gain). Line 10 of Alg. 5 is the major change to Round-Robin Greedy specifically for the cardinality constraint case, where we first find the solution to the cardinality constrained submodular

welfare problem $\max_{\pi \in \Pi(V', m', k)} \sum_{S \in \pi} f(S)$, and then only apply Round-Robin Greedy to the union $V''$ of the solution $A'_1, A'_2, \ldots, A'_{m'}$.

The running time of Alg. 5 is similar to Round-Robin Greedy, with additional costs caused by Line 10, which solves a cardinality constrained submodular welfare problem. Using Continuous Greedy and swap rounding [7] for Line 10 can be quite costly ($\mathcal{O}(n^5)$ for the inner loop), which may improve in the future by a better algorithm. In Alg. 7, we propose another algorithm that addresses the constrained robust submodular problem with $\mathcal{C}$ as any matroid constraint $\mathcal{M}$ and incurs no additional computation costs compared to Round-Robin Greedy.

**Theorem 2 (Matroid Constrained Round-Robin).** *For the problem in Eq. (4), with $\mathcal{C}$ as any matroid constraint $\mathcal{M}$, Alg. 7 gives a solution $\min_{j \in [m]} f(A_j) \geq \frac{(1-e^{-1})}{5} \min_{j \in [m]} f(O_j^{\mathcal{M}})$. The time complexity of of Alg. 7 is $\mathcal{O}(n^2 (\log \log m + \log \frac{1}{\delta}))$.*

---

**Algorithm 7:** Matroid Round-Robin Greedy

---

**input** : $f, V, m$, matroid constraint $\mathcal{M}$, discounting factor for guessing optimal $\delta$

1 Let $\tau$ be the solution value of Alg. 3;
2 Let $high = \lceil \log_{1+\delta}(m+2) \rceil$, $low = 0$;
3 Create a sequence of guessed values: $(\tau, (1+\delta)\tau, (1+\delta)^2 \tau, \ldots, (1+\delta)^{high} \tau)$;
4 Create an empty solution ($\emptyset$ for each block in the partition) for each guessed value $\pi_0, \pi_1, \ldots, \pi_{high}$;
5 **while** $high \geq low$ **do**
6     Let $idx = \lfloor (high + low)/2 \rfloor$; Let $A_1 = A_2 = \ldots = A_m = \emptyset$;
7     Let $V' = \{v | v \in V, f(v) \leq \frac{1-e^{-1}}{5}(1+\delta)^{idx} \tau\}$; Let $G = V \setminus V'$;
8     Assign $G$ to $A_{m-|G|+1}, A_{m-|G|+2}, \ldots, A_m$ with one element per block;
9     Let $m' = m - |G|$;
10     Let $\{A_1, A_2, \ldots, A_{m'}\} = RR(f, V', m', \mathcal{M}, [m'])$;
11     **if** $f(A_j) \geq \frac{(1-e^{-1})}{5}(1+\delta)^{idx} \tau \; \forall j \in [m']$ **then**
12        Let $\pi_{idx} = \{A_1, A_2, \ldots, A_m\}$; Let $low = idx + 1$;
13     **else**
14        Let $high = idx - 1$;
15 **return** best of $\pi_0, \pi_1, \ldots, \pi_{high}$;

---

Comparing to Alg. 5, the major change in Alg. 7 is (1) we do not need to run the costly Continuous Greedy and swap rounding to get a solution to the constrained welfare problem, which makes the algorithm applicable in practice; (2) for the $RR(\cdot)$ subroutine, we find a feasible element and add it to the current block, and we remove a block from the candidate set $J$ if there are no element in the remaining set that can be added to the block without violating the matroid constraint. Note for the cardinality constraint case, we can always find a feasible element until every block has $k$ elements. The overall running time is $\mathcal{O}(n^2 (\log \log m + \log \frac{1}{\delta}))$. The general idea of proving Theorem 2 is to bound the solution to the fractional solution of the continuous relaxation. For every block in the solution, we inspect the elements that have been evaluated during the greedy step. For those elements with large gains when being evaluated but not added due to the violation of the matroid constraint, we bound their gains as submodular maximization with a matroid constraint on a reduced ground set. For the remaining elements, we bound their gains by the greedy step and together we get the desired bound.

## 6 Experiments

We empirically test Algs. 3 and. 7 on the CIFAR-10 training set [19] ($|V| = 50000$). We use facility location as our submodular function, i.e., $f(S) = \sum_{v \in V} \max_{v' \in S} sim(v, v')$, where $sim(v, v')$ measures the affinity between elements $v$ and $v'$. This function is widely used and naturally describes a subset's diversity via its similarities/distances to all other points in the ground set, and it has achieved much practical success [27, 26, 23]. For similarity, we use a Gaussian kernel with L2 distances, i.e., $sim(v, v') = exp(\frac{-||v-v'||_2}{\sigma})$, where $\sigma$ is the bandwidth of the kernel, set to the average L2 distance, i.e., $\sigma = \sum_{v, v' \in V} ||v - v'||_2 / n^2$. The features used to calculate the L2 distance is the bottleneck layer's outputs generated by a deep auto-encoder model (details are in Appendix D).

We test the algorithms and compare their objective values (Eq. (4)) with a matroid constraint where we limit the number of samples selected for each class in CIFAR-10 for each block (CIFAR-10 has 10 classes). We compare the two algorithms with a random selection baseline — we randomly sample from each class and assign to each block with the constrained number of elements. Hence, the random selection results satisfy the constraints. In Fig. 1, we report the results for different matroid constraints with various block sizes. The random baseline results are reported with means and standard-deviations over 10 runs. For all cases, we see that both Alg. 3 and Alg. 7 significantly outperform the baselines. Although Alg. 7 has a better theoretical bound, Alg. 3 consistently gives better performance. Intuitively, Alg. 3 directly optimizes the objective as it greedily adds elements to the minimum-valued block. We expect Alg. 3 to perform better in practice compared to Alg. 7 as Alg. 7 has a fixed ordering of the blocks, and the minimum-valued block tends to be the last block in the ordering, in which case it does not get to select samples that have already been selected by prior blocks in the ordering. In Fig. 2, we use the partitioned blocks as minibatches to train a ResNet-9 model and compare their performance on the

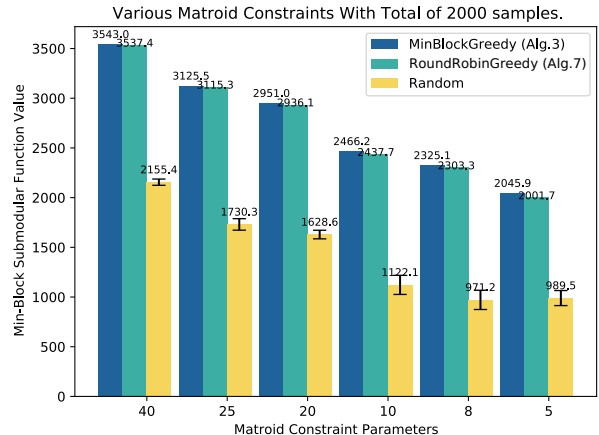

**Figure 1:** Matroid Constrained results. The $x$-axis denotes the number of samples per class for the matroid. A total of 2000 samples are selected for each case.

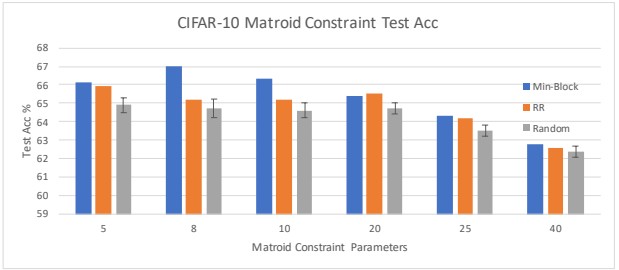

**Figure 2:** Training ResNet-9 (Myrtle AI) on partitioned mini-batches under matroid constraints with various parameters. The $y$-axis denotes the test set accuracy. The $x$-axis denotes the number of samples per class for the matroid.

test set. We observe that the blocks with higher submodular evaluations tend to generate better performance for the trained model. We also provide results on synthetic data in Appendix Section C.

## 7 Conclusions

We study the problem of constrained submodular robust partitioning. We propose two classes of algorithms, Min-Block Greedy and Round-Robin Greedy based, and prove approximation bounds under various constraints. This improves the applicability of the robust partitioning framework to different scenarios. In future work, we wish to extend the current approach to the heterogeneous submodular partitioning setting where each block may be evaluated by a different submodular function. Given the good performance of Alg. 3 in practice, it is worth investigating if further conditions or modifications to Alg. 3 yield improved theoretical bounds.

This work was supported in part by the CONIX Research Center, one of six centers in JUMP, a Semiconductor Research Corporation (SRC) program sponsored by DARPA.

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
