We restate the theoretical statements and the algorithms here for completeness and convenience.

# A  Proofs for Minimum Block Greedy Based Algorithm

Given a normalized monotone submodular function $f : 2^V \to \mathbb{R}_{\geq 0}$, where $V$ is the ground set with $|V| = n$ elements, and a positive integer $m$ denoting the number of blocks in the partition, the Submodular Robust partition is defined as:

$$\max_{\pi \in \Pi(V,m)} \min_{A \in \pi} f(A), \tag{1}$$

Where $\Pi(V,m)$ is the set of all possible partitions on set $V$ into $m$ blocks, and $\pi$ is one partition. The objective aims to find the partition such that the minimum-valued block in the partition is maximized. Denote the optimal partition to Eq. (1) as $\pi^* = \{O_1, O_2, \dots, O_m\}$.

**Lemma 2 (Unconstrained Min-Block Streaming[29]).** *For a ground set $V$ and its elements $(v_1, v_2, \dots, v_n)$ coming in an arbitrary streaming order, the output solution of Alg. 1 has $\min_{j \in [m]} f(A_j) \geq \frac{1}{m} \min_{j \in [m]} f(O_j)$.*

**Corollary 1 (Unconstrained Min-Block Greedy[29]).** *The output solution of Alg. 2 has $\min_{j \in [m]} f(A_j) \geq \frac{1}{m} \min_{j \in [m]} f(O_j)$ since the order of adding elements in Min-Block Greedy is one possible order of the ground set elements.*

**Lemma 3 (Tightness of Corollary 1 ).** *$\forall \epsilon > 0$, $\exists$ a submodular function $f$ such that the output solution of Alg 2 $\min_{j=1:m} f(A_j) = \frac{1}{m} \min_{j=1:m} f(O_j) + \epsilon$.*

*Proof.* We construct a set cover function as the tight example for Corollary 1. We illustrate the set cover function graphically in Fig. 3.

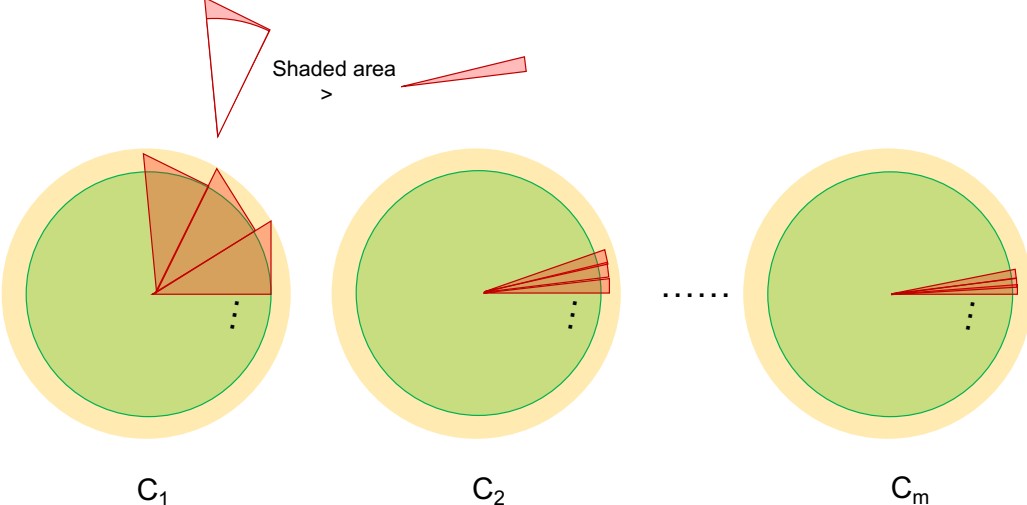

**Figure 3:** A graphical illustration of the tight example. The circles are the areas to cover for the set cover function and the green inner circles and the red triangles are elements in the ground set (the outer yellow circles are not elements). The inner circles (green) largely overlap with the outer circles (yellow). The red triangles mostly overlap with the inner circle, with little gains on the ring between the two circles. We can change the size of the red triangles so that Min-Block Greedy prefers a redundant element (the shaded area comparison on the top of the figure). Also note that the red triangles may overlap on the inner circle part (they may not retain the shapes as triangles), so overall they cover $m - 1$ times the area of each circle.

Suppose we have $m$ circles of area to cover in the set cover function. Say we order the circles by their area, say $C_1 < C_2 < \dots < C_m$. Let $C_{j+1} = C_j + \epsilon_{j+1}$ for some $\epsilon_{j+1} > 0$. For every circle $j$, we have an element $v_j \in V$, which covers an inner circle, which almost covers the entire circle. W.l.o.g, suppose $f(v_1) = 1$ and let $C_j = f(v_j) + \delta_j$.

For each circle $C_j$, we construct $n_j$ elements, which largely overlap with the inner circle covered by $v_j$ and gives little gain on the ring between the inner circle and the outer circle $C_j$. Call these $n_j$ elements $V_j$. Let $f(V_j|v_j) = \epsilon'_j$, $f(v) < f(v_j)\forall v \in V_j$, and $f(V_j) > f(v_j)$.

Now let's focus on the first two circles $C_1$ and $C_2$, and assume $m = 2$ for the partition problem. It is easy to extend to general $m$ case by recursively applying the following arguments on $C_2$ and $C_3$.

Suppose we run the min-block greedy, after the first two steps, one block contains $v_1$ and the other contains $v_2$. At step 3, $v_1$ is the min-block. By setting the suitable values for $n_1$ and $n_2$ (say $n_2 >> n_1$), we can make $f(v|v_1) > f(v'|v_1)\forall v \in V_1, v' \in V_2$. Therefore, we will still select an element from $V_1$ even though such element overlaps largely with the inner circle $v_1$. We can force the min-block greedy algorithm to select all elements from $V_1$ before the min-block changes to the block containing $v_2$, and $f(V_1 \cup v_1) > f(v_2)$. After that, the algorithm can only add elements from $V_2$ to the block containing $v_2$, which gives only $\epsilon'_2$ gains. As we can make the values of $\epsilon_j, \epsilon'_j$ and $\delta_j$ arbitrarily small, the solution is arbitrarily close to 1. On the contrary, the optimal partition should add elements in $V_1$ to $v_2$ and elements in $V_2$ to $v_1$, and the solution has value arbitrarily close to 2.

To extend to general $m$ partitions, we may treat the current $V_2$ as $V_1$, and construct $V_3$ in the same way we construct $V_2$ based on $V_1$. In the first $m$ steps of the min-block greedy, the algorithm is forced to evenly distribute $v_j, j = 1, 2, \ldots, m$ into every block. After that, the algorithm adds all elements in $V_j$ to the block containing $v_j$ before the min-block changes and the block containing $v_j$ will not become the min-block again. In the end, every block only covers (almost) one circle. Suppose for $V_j$, we may make the elements to cover the inner circle multiple times, i.e., $\exists \pi \in \Pi(V_j, m)s.t.\forall A \in \pi, f(A) \geq f(v_j)$. Then for the optimal solution, every block can cover (almost) all the circles, and therefore the approximation ratio can be arbitrarily close to $1/m$.

□

The constrained submodular robust partition problem:

$$\max_{\pi \in \Pi(V,m,\mathcal{C})} \min_{A \in \pi} f(A), \tag{4}$$

Where $\Pi(V, m, \mathcal{C})$ is the set of all possible partitions on set $V$ into $m$ blocks such that for every partition $\pi \in \Pi(V, m, \mathcal{C})$, every block $A \in \pi$ should satisfy the constraint $A \in \mathcal{C}$. Denote the optimal partition to Eq. (4) as $\pi^*_C = \{O^{\mathcal{C}}_1, O^{\mathcal{C}}_2, \ldots, O^{\mathcal{C}}_m\}$.

**Theorem 1 (Constrained Min-block Greedy).** *Given a constraint $\mathcal{C}$, if the greedy solution $S^g$ to problem $\max_{S \in \mathcal{C}} f(S)$ using Alg. 4 has a bound of $\alpha$, i.e., $f(S^g) \geq \alpha \max_{S \in \mathcal{C}} f(S)$, then the solution of Alg. 3 has $\min_{j \in [m]} f(A_j) \geq \frac{\alpha}{\alpha m + 1} \min_{j \in [m]} f(O^{\mathcal{C}}_j)$. Assuming GreedyStep($\cdot$) takes $\mathcal{O}(1)$ oracle calls, the time complexity of of Alg. 4 is $\mathcal{O}(n^2)$.*

*Proof.* W.l.o.g., we assume that the block of index 1 for a partition corresponds to the minimum-valued block, e.g., $f(O^{\mathcal{C}}_1) = OPT^{\mathcal{C}}$. For Min-Block Greedy algorithm, we always add an element feasible to the constraint $\mathcal{C}$ to the block with the minimum evaluation. Let the minimum block in our final solution be $A_1$. Due to the final singleton comparison step (line 9-11 in Alg. 3), there are several different scenarios for $A_1$:

1. It is never the case that we cannot add any elements to a block due to the constraint (line 5 always true). This is the simplest case as we can directly reduce it to a stream of elements with the same ordering as we add them into different blocks, and Lemma. 2 applies. We therefore can get an $1/m$ approximation ratio, which is better than the one given in the theorem for any $\alpha \leq 1$.

2. $A_1$ is the first block that we cannot find any feasible elements to add. The singleton comparison step may increase the function value of $A_1$. however, by assumption it's still the minimum block after the algorithm completes.

3. There are other blocks that we cannot find any feasible elements to add before $A_1$. This could only happen if the other blocks get their values increased by the singleton comparison step. As if the singleton comparison step does not swap the block with the largest singleton, the block, which is not $A_1$ in this case, is the minimum block for that step and remains minimum for the following steps of the algorithm.

For scenarios 2 and 3, the general idea of the proof is the same, where we separate the ground set $V$ into two parts $V'$ and $R'$ ($V = V' \cup R'$), and bound $f(A_1)$ by comparing to a block in the optimal solution $O_j^\mathcal{C}$ through $f(O_j^\mathcal{C} \cap V')$ and $f(O_j^\mathcal{C} \cap R')$. However, for 2 and 3, we will use slightly different $V'$ and $R'$.

First, for scenario 2), let's suppose at step $t'$, the current minimum block is $A_1$, and we find no feasible elements to add. Let all the elements allocated so far (before the singleton comparison step for $A_1$) as $V'$, and the remaining unallocated elements as $R'$. $V = V' \cup R'$. Denote the elements in $A_1$ before the singleton comparison step as $A_1'$, as the singleton step always improves the block value, we have $f(A_1) \geq f(A_1')$.

If we run the min-block robust partition greedy algorithm on $V'$ only, we will get the same partial partition as we run on $V$ for $t'$ steps. Therefore, suppose we create a stream that orders the elements in $V'$ in the same order that those elements get allocated by the min-block robust partition greedy algorithm, then by Lemma. 2, we have:

$$f(A_1) \geq f(A_1') \geq \frac{1}{m} OPT(V'),\tag{7}$$

Where we denote $OPT(V') = \max_{\pi \in \Pi(V',m)} \min_{A \in \pi} f(A)$ as the optimal solution for the unconstrained robust submodular partition on the ground set $V'$.

Let $O_j^\mathcal{C}$ be some block in the optimal constrained partition on ground set $V$. since $O_j^\mathcal{C}$ can be the non-minimal block in the optimal solution, we have:

$$f(O_j^\mathcal{C}) \geq OPT^\mathcal{C}.\tag{8}$$

There exists a $j \in \{1, \ldots, m\}$ such that

$$f(A_1) \geq \frac{1}{m} OPT(V')\tag{9}$$

$$\geq \frac{1}{m} f(O_j^\mathcal{C} \cap V'),\tag{10}$$

as otherwise $\forall j \in \{1, \ldots, m\}, O_j^\mathcal{C} \cap V'$ forms a solution for the partition problem on the reduced ground set $V'$, and gives a solution value better than $OPT(V')$, which violates the optimality of $OPT(V')$.

Now we separate the constrained optimal solution on ground set $V$ into 2 parts: $O_j^\mathcal{C} \cap V'$ and $O_j^\mathcal{C} \cap R'$.

**Assumption 1**. Suppose

$$f(O_j^\mathcal{C} \cap R') \geq f(O_j^\mathcal{C} \cap V'),\tag{11}$$

Then because of submodularity, $f(O_j^\mathcal{C} \cap R') + f(O_j^\mathcal{C} \cap V') \geq f(O_j^\mathcal{C})$ (recall $V = V' \cup R'$ )and we have

$$f(O_j^\mathcal{C} \cap R') \geq \frac{1}{2} f(O_j^\mathcal{C}).\tag{12}$$

Consider the set $R' \cup A_1'$, let

$$\hat{O} \in \underset{S \subseteq R' \cup A_1', S \in \mathcal{C}}{\operatorname{argmax}} f(S),\tag{13}$$

I.e., $\hat{O}$ is the optimal solution to the constraint submodular max on the reduced ground set $R' \cup A_1'$. After the singleton comparison step on $A_1'$, we get $A_1$, which is the greedy solution of Alg. 3 on the reduced ground set $R' \cup A_1'$ and constraint $\mathcal{C}$. Therefore, based on the $\alpha$-bound assumption in Theorem 1, we have:

$$f(A_1) \geq \alpha f(\hat{O})\tag{14}$$

$$\geq \alpha f(O_j^\mathcal{C} \cap R')\tag{15}$$

$$\geq \frac{\alpha}{2} f(O_j^\mathcal{C})\tag{16}$$

$$\geq \frac{\alpha}{2} f(O_1^\mathcal{C}).\tag{17}$$

Eq. (15) comes from the optimality of $\hat{O}$ and Eq. (16) comes from **Assumption 1**.

**Assumption 2**. Otherwise, we have

$$f(O_j^{\mathcal{C}} \cap V') > f(O_j^{\mathcal{C}} \cap R') \tag{18}$$

$$\geq \frac{1}{2}f(O_j^{\mathcal{C}}). \tag{19}$$

We therefore have:

$$f(A_1) \geq \frac{1}{m}f(O_j^{\mathcal{C}} \cap V') \tag{20}$$

$$> \frac{1}{2m}f(O_j^{\mathcal{C}}) \tag{21}$$

$$\geq \frac{1}{2m}f(O_1^{\mathcal{C}}) \tag{22}$$

Note that one of **Assumption 1** and **Assumption 2** is always true, since $f(O_j^{\mathcal{C}} \cap R') + f(O_j^{\mathcal{C}} \cap V') \geq f(O_j^{\mathcal{C}})$ because of submodularity. Previously, we use equal weights of $\frac{1}{2}$ for both assumptions. We can balance the weights as long as the weights sum to one, and we get:

if $f(O_j^{\mathcal{C}} \cap R') \geq \frac{1}{\alpha m+1}f(O_j^{\mathcal{C}})$, we have

$$f(A_1) \geq \alpha f(\hat{O}) \tag{23}$$

$$\geq \alpha f(O_j^{\mathcal{C}} \cap R') \tag{24}$$

$$\geq \frac{\alpha}{\alpha m + 1}f(O_j^{\mathcal{C}}) \tag{25}$$

$$\geq \frac{\alpha}{\alpha m + 1}f(O_1^{\mathcal{C}}); \tag{26}$$

if $f(O_j^{\mathcal{C}} \cap V') > \frac{\alpha m}{\alpha m+1}f(O_j^{\mathcal{C}})$, we have

$$f(A_1) > \frac{1}{m}f(O_j^{\mathcal{C}} \cap V') \tag{27}$$

$$> \frac{1}{m}\frac{\alpha m}{\alpha m + 1}f(O_j^{\mathcal{C}} \cap V) \tag{28}$$

$$> \frac{\alpha}{\alpha m + 1}f(O_1^{\mathcal{C}}). \tag{29}$$

Thus, we get a $\frac{\alpha}{\alpha m+1}$ bound.

For scenario 3, we only need to change $V'$ and $R'$ and the same argument follows. Recall that in such a scenario, there are some other blocks that have no feasible elements to add before $A_1$, and they get their values increased through the singleton comparison step. There are also two different cases here. Firstly, it does not happen that there are no feasible elements to add to $A_1$ until the end of the algorithm. In such a case, similar to the scenario 1, we can order the elements as a stream and applies Lemma. 2 to get the $1/m$ approximation ratio. Note that the blocks that get to the singleton comparison step all get their values increased for scenario 3, and we can just add the singleton $a^*$ (line 9) to that block in the streaming case. To be more precise, for Alg. 3 when block $j$ ($j \neq 1$) is the current minimum block, and has no feasible elements to add, we denote its elements as $A_j'$, and the singleton comparison step gives an element $a^*$ with $f(\{a^*\}) \geq f(A_j')$. In the streaming ordering, we use the same ordering as we add element in Alg. 3, and at the singleton comparison step for block $j$, we have the next element in the stream be $a^*$, and we add that element to block $j$ since block $j$ is the current minimum block. By monotonicity, we have $f(A_j' \cup \{a^*\}) \geq f(\{a^*\}) \geq f(A_1)$. In other words, those blocks never become the minimum block again, and no elements get added to them after their singleton comparison step. Therefore, we have a streaming ordering of the elements that will make the minimum block equal to $A_1$ and Lemma. 2 applies.

Next, we discuss for the case where it happens that there are no feasible elements to add to $A_1$. When that happens, we set all the allocated elements as $V'$ and the remaining elements as $R'$ before the singleton comparison step. Note for those blocks that get to the singleton comparison step before $A_1$, we will also include the singletons in $V'$ Recall those singletons have larger gains and get swapped

with the elements in those blocks for this scenario. As stated above, those blocks with the singleton comparison step never become the minimum block again, so they don't interfere with the remaining blocks/elements. For such $V'$ and $R'$, the exact argument in scenario 2 can be made, i.e., we can treat $A_1'$ as a min-block streaming solution on $V'$ (Eq. 20) and $A_1$ as a greedy solution on $A_1' \cup R'$ (Eq. 15). $\qquad\square$

**Corollary 2 (Cardinality Constrained Min-block Greedy).** *For $\mathcal{C}$ as a cardinality constraint, the output of Alg 3 has* $\min_{j=1:m} f(A_j) \geq \frac{1}{m + \frac{1}{1-e^{-1}}} \min_{j=1:m} f(O_j^{\mathcal{C}})$.

**Corollary 3 (Matroid Constrained Min-block Greedy).** *For $\mathcal{C}$ as an intersection of $p$ matroids constraint, the output of Alg 3 has* $\min_{j=1:m} f(A_j) \geq \frac{1}{m+p+1} \min_{j=1:m} f(O_j^{\mathcal{C}})$.

**Corollary 4 (Knapsack Constrained Min-block Greedy).** *For $\mathcal{C}$ as a knapsack constraint, the output of Alg 3 has* $\min_{j=1:m} f(A_j) \geq \frac{1}{m + \frac{2}{1-1/e}} \min_{j=1:m} f(O_j^{\mathcal{C}})$.

# B    Proofs for Round-Robin Greedy Based Algorithms

The matroid constrained submodular robust partition problem is

$$\max_{\pi \in \Pi(V,m,\mathcal{M})} \min_{A \in \pi} f(A). \tag{30}$$

Before we get into the proofs for the algorithm bounds, we will state the following lemma, which is a general property about robust submodular partitioning.

**Lemma 5 (Removal of one element and one block).** *For any $v \in V$, we have:*

$$\max_{\pi \in \Pi(V \setminus v, m-1, \mathcal{M})} \min_{A \in \pi} f(A) \geq \max_{\pi \in \Pi(V,m,\mathcal{M})} \min_{A \in \pi} f(A). \tag{31}$$

*I.e., if we remove one element and one block from the problem, the optimal solution gets no worse.*

*Proof.* Denote the optimal solution on $V$ and $m$ by $O_1, O_2, \ldots, O_m$ with $f(O_1) \leq f(O_2) \leq \ldots \leq f(O_m)$.

Suppose $v \in O_j$ for some $j$, then the blocks other than $O_j$ forms a solution for problem defined on $V \setminus v$ and $m - 1$, and we can add elements in $O_j \setminus v$ to other blocks (if the constraints permit). In the worst case, even if we cannot add any elements of $O_j \setminus v$ to other blocks, we still have $\max_{\pi \in \Pi(V,m,\mathcal{M})} \min_{A \in \pi} f(A) \geq \min_{j' \in [m], j' \neq j} f(O_{j'}) \geq f(O_1)$.

Suppose $\forall j \in [m]$, $v \notin O_j$, then we only remove one block, and we can add the elements in that block to any other block so the solution value gets improved. $\qquad\square$

For Round-Robin Greedy based algorithms, we first guess the optimal solution value and then assign singletons to blocks which satisfies the bound based on the guessed optimal value. After that, we run the algorithm on the restricted problem with those blocks and elements removed. By applying the previous lemma (recursively if multiple elements and blocks removed), we know that the optimal solution on the restricted instance is no worse than the optimal solution on the original problem. Therefore, it suffices to analyze the solution on the restricted instance.

**Lemma 4 (Cardinality Constrained Round-Robin).** *For the problem in Eq. (4), with $\mathcal{C}$ as a cardinality constraint $k$, Alg. 5 gives a solution* $\min_{j \in [m]} f(A_j) \geq \frac{(1-e^{-1})^2}{3} \min_{j \in [m]} f(O_j^k)$.

*Proof.* By solving $\max_{\pi \in \Pi(V',m',k)} \sum_{S \in \pi} f(S)$ in Line 10 of Alg. 5 (Theorem III.3 in [7]), we know that

$$\sum_{j \in [m]'} f(A_j') \geq (1-e^{-1}) \max_{\pi \in \Pi(V',m',k)} \sum_{S \in \pi} f(S). \tag{32}$$

---

**Algorithm 5:** Cardinality Round-Robin Greedy

---

**input** : $f, V, m$, cardinality constraint $k$, discounting factor for guessing optimal $\delta$

1 Let $\tau$ be the solution value of Alg. 3;
2 Let $high = \lceil \log_{1+\delta}(m + 2) \rceil$, $low = 0$;
3 Create a sequence of guessed values: $(\tau, (1 + \delta)\tau, (1 + \delta)^2\tau, \ldots, (1 + \delta)^{high}\tau)$;
4 Create an empty solution ($\emptyset$ for each block in the partition) for each guessed value
  $\pi_0, \pi_1, \ldots, \pi_{high}$;
5 **while** $high \geq low$ **do**
6  $\quad$ Let $idx = \lfloor (high + low)/2 \rfloor$; Let $A_1 = A_2 = \ldots = A_m = \emptyset$;
7  $\quad$ Let $V' = \{v | v \in V, f(v) \leq \frac{(1-e^{-1})^2}{3}(1 + \delta)^{idx}\tau\}$; Let $G = V \setminus V'$;
8  $\quad$ Assign $G$ to $A_{m-|G|+1}, A_{m-|G|+2}, \ldots, A_m$ with one element per block;
9  $\quad$ Let $m' = m - |G|$;
10 $\quad$ Let $A'_1, A'_2, \ldots, A'_{m'}$ be the solution to $\max_{\pi \in \Pi(V', m', k)} \sum_{S \in \pi} f(S)$ using continuous
    $\quad$ greedy and swap rounding; Let $V'' = \cup_{j \in [m]'} A'_j$;
11 $\quad$ Let $\{A_1, A_2, \ldots, A_{m'}\} = RR(f, V'', m', \mathcal{M}_k^u, [m'])$;
12 $\quad$ **if** $f(A_j) \geq \frac{(1-e^{-1})^2}{3}(1 + \delta)^{idx}\tau \; \forall j \in [m']$ **then**
13 $\quad\quad$ Let $\pi_{idx} = \{A_1, A_2, \ldots, A_m\}$; Let $low = idx + 1$;
14 $\quad$ **else**
15 $\quad\quad$ Let $high = idx - 1$;
16 **return** best of $\pi_0, \pi_1, \ldots, \pi_{high}$;

---

Recall that we denote the optimal solution value in the cardinality constraint case by $OPT^{\mathcal{M}_k^u}$, where $k$ is the cardinality. We assume we know the optimal solution value $OPT^{\mathcal{M}_k^u}$ for this proof. For the algorithm, the $OPT^{\mathcal{M}_k^u}$ value is guessed within a factor of $\frac{1}{1+\delta}$. Therefore, to be more precise, we have an additional factor of $\frac{1}{1+\delta}$ in the bound, which can be made arbitrarily small by setting $\delta$ small.

For the limited ground set $V''$, running unconstrained round-robin ensures every block to have at most $k$ elements and therefore the cardinality constraint is satisfied. Suppose we run the continuous greedy algorithm on the limited ground set $V''$ with the submodular welfare objective ($\max_{\pi \in \Pi(V'', m')} \sum_{S \in \pi} f(S)$), we get a fractional solution $x_1 = x_2 = \ldots = x_{m'} = (\frac{1}{m'}, \frac{1}{m'}, \ldots, \frac{1}{m'})$ (we do not really need to run the algorithm, but we will compare our solution to the fractional solution). Denote the multilinear extension of $f$ by $F$ and $F(x) = \mathbb{E}_{R \sim x} f(R)$ (we can think it as the expected value of $f$ where every element is sampled independently based on probabilities defined in vector $x$). Consider any block $A_j$ in the solution of Alg 5 for $j \in [m']$ (for $j \notin [m']$, those blocks are the singleton assignment blocks and they satisfy the bound by construction), we have:

$$\frac{(1-e^{-1})^2}{3}OPT^{\mathcal{M}_k^u} + 2f(A_j) \geq F(x_j) \tag{33}$$

$$\geq \frac{1-e^{-1}}{m'} \max_{\pi \in \Pi(V'', m')} \sum_{S \in \pi} f(S) \tag{34}$$

$$\geq \frac{1-e^{-1}}{m'} \max_{\pi \in \Pi(V'', m', k)} \sum_{S \in \pi} f(S) \tag{35}$$

$$\geq \frac{(1-e^{-1})^2}{m'} \max_{\pi \in \Pi(V', m', k)} \sum_{S \in \pi} f(S) \tag{36}$$

$$\geq (1-e^{-1})^2 \max_{\pi \in \Pi(V', m', k)} \min_{S \in \pi} f(S). \tag{37}$$

$$\geq (1-e^{-1})^2 OPT^{\mathcal{M}_k^u}. \tag{38}$$

Rearrange and we get:

$$f(A_j) \geq \frac{(1-e^{-1})^2}{3}OPT^{\mathcal{M}_k^u}. \tag{39}$$

Eq. (33) comes from the Lemma.3 of [3], in which case we can bound every block in the round-robin solution to the fractional solution of the continuous greedy algorithm on the multilinear extension of $f$. Note for Lemma.3 of [3], they study the unconstrained case and show that $\frac{\gamma}{3}OPT^{\mathcal{M}_k^u} + 2f(A_j) \geq F(x_j)$ with $\gamma = 1 - e^{-1}$. $\gamma$ comes from the singleton assignment step, which assigns blocks with singletons whose values are larger than $\frac{\gamma}{3}OPT^{\mathcal{M}_k^u}$. A slightly more general statement can be made for any $0 \leq \gamma \leq 1$ with the same proof as Lemma.3 of [3]. In our case, we pick $\gamma = (1 - e^{-1})^2$. Eq. (34) follows from the property of the continuous greedy solution. The continuous greedy gives a $(1 - e^{-1})$ approximation to the submodular welfare problem, and the fractional solution $x_j$ for each block is the same. Therefore, every block's evaluation in expectation is at least $\frac{(1-e^{-1})}{m'}$ of the submodular welfare optimal solution. Eq. (35) follows that the unconstrained solution is no worse than the constrained solution. Eq. (36) uses Eq. (32): $A_1', \ldots, A_m'$ is one possible solution to $\max_{\pi \in \Pi(V'', m', \mathcal{M}_k^u)} \sum_{S \in \pi} f(S)$, and we know $\sum_j f(A_j') \leq \max_{\pi \in \Pi(V'', m', k)} \sum_{S \in \pi} f(S)$ because of the $\max$ operator. Therefore, we have $\max_{\pi \in \Pi(V'', m', k)} \sum_{S \in \pi} f(S) \geq \sum_j f(A_j') \geq (1 - e^{-1}) \max_{\pi \in \Pi(V', m', k)} \sum_{S \in \pi} f(S)$. Eq. (37) follows that the sum over blocks of the max-min solution is no larger than the optimal welfare solution. Eq. (38) uses Lemma. 5: the optimal max-min solution on $V'$ and $m'$ is no worse than the optimal max-min solution on $V$ and $m$.

As stated above, $\frac{\gamma}{3}OPT + 2f(A_j) \geq F(x_j)$ is true for any $0 \leq \gamma \leq 1$. It may seem that making $\gamma$ smaller can improve the bound. However, we only bound the $m'$ blocks but not the singleton blocks. Because of the singleton assignment step, every singleton has a value larger than $\frac{\gamma}{3}$, and the final bound over all $m$ blocks will be the minimal of the bound on the $m'$ blocks and the singleton blocks. Setting $\gamma$ small worsens the bound on the singleton blocks. To balance the two bounds, $\gamma = (1 - e^{-1})^2$ is picked so that the bounds on the $m'$ blocks and the singleton blocks meet.

Finally, we discuss about the binary search process. Let's consider a special problem instance, which does not have any singleton values larger than $\frac{(1-1/e)^2}{3}$ times the optimal solution value of the problem instance. Then we can directly run line 10 and 11 to get a solution with a $\frac{(1-1/e)^2}{3}$ approximation ratio (no optimal solution value guessing and large singleton value removal). However, we don't know if that assumption is true for general problem instances.

Back to the general problem instances. Suppose the guessed optimal values form a sequence $(\tau_1, \tau_2, \ldots, \tau_l)$ where $\tau_{i+1} = (1 + \delta)\tau_i$. We are going to show that for any $\tau_i \leq OPT^{\mathcal{M}_k^u}$ as the guessed optimal value that we plug into the binary search iterations (Alg. 5 line 7-15), line 12 of Alg. 5 is always true. For simplicity, denote the optimal solution of the cardinality constrained robust partitioning problem as $OPT$ for the following. Let's denote the found large singleton values and the remaining sets (line 7 of Alg. 5) respectively by $G_{OPT}$ and $V_{OPT}'$ for $OPT$, and $G_i$ and $V_i'$ for $\tau_i$. Since $\tau_i \leq OPT$, $G_{OPT} \subseteq G_i$ as the threshold is smaller for $\tau_i$. Then by Lemma. 5, we know that the optimal solution $OPT_i$ on $V_i'$ (partitioned to $m - |G_i|$ blocks) is no less than the optimal solution $OPT'$ on $V_{OPT}'$ (partitioned to $m - |G_{OPT}|$ blocks). Also note that since we remove all singleton values to form $V_i'$ based on the threshold $\frac{(1-e^{-1})^2}{3}\tau_i$, and $\tau_i \leq OPT \leq OPT' \leq OPT_i$ ($OPT \leq OPT'$ is also from Lemma. 5), we are guaranteed that there are no singleton elements with $f(v) \geq \frac{(1-e^{-1})^2}{3}OPT_i$ in $V_i'$. Therefore, as discussed for the special problem instance, line 10 and 11 on $V_i'$ give a solution whose every block has a value of at least $\frac{(1-e^{-1})^2}{3}OPT_i \geq \frac{(1-e^{-1})^2}{3}\tau_i$, and thus line 12 is guaranteed to be true for $\tau_i$.

Based on that, either of the following must hold: 1) for all $\tau_i > OPT^{\mathcal{M}}$ line 12 is false, and in such a case, we can use binary search to find the largest $\tau_i$ with line 12 true, and let's call it $\tau_{i^*}$; 2) there exists some $\tau_i > OPT^{\mathcal{M}}$ such that line 12 is true. If we find such $\tau_i$, we find a solution with $f(A_j) \geq \frac{(1-e^{-1})^2}{3}\tau_i \geq \frac{(1-e^{-1})^2}{3}OPT^{\mathcal{M}_k^u}$. Otherwise if we don't find it, we go to the first case and will still find $\tau_{i^*}$. We can therefore conclude that binary search can be applied to search among the guessed optimal solution values.

$\square$

## B.1 Proof for the Matroid Constraint Case

**Lemma 6 (Continuous Greedy Solution).** *For the constrained welfare problem $\max_{\pi \in \Pi(V, m, \mathcal{M})} \sum_{A \in \pi} f(A)$, the continuous greedy algorithm outputs a fractional solution $x_1 = x_2 = \ldots = x_m$ ($x_j \in [0,1]^n$), which is the same for every block in the partition and $\sum_{j \in [m]} F(x_j) \geq (1 - e^{-1}) \max_{\pi \in \Pi(V, m, \mathcal{M})} \sum_{A \in \pi} f(A)$. $F$ is the multilinear extension of $f$, i.e., $F(x) = \mathbb{E}_{R \sim x} f(R)$ (we can think it as the expected value of $f$ where every element is sampled independently based on probabilities defined in vector $x$). Moreover, $\forall i \in [n], j \in \{1, \ldots, m\}, x_j[i] \leq \frac{1}{m}$ and $\sum_{i \in V} x_j[i] \leq r_{\mathcal{M}}(V)$.*

*Proof.* Note that the continuous greedy can give a fractional solution with $1 - e^{-1}$ bound under any solvable polytope constraint. It's the rounding procedure that limits the constraint we can use to get a set solution, e.g., with pipage rounding, we can use any matroid constraint.

In fact, we do not need to run the continuous greedy algorithm, and we only need to show the existence of a solution. Suppose the solution to the $max_y\{w \cdot y, y \in \mathcal{P}\}$ step of the continuous greedy algorithm is given by some oracle. Given the direction $y$, we just evenly split the resulting vector $y$ among the $m$ blocks, as we cannot distinguish between blocks. At the end of the algorithm, we will have the fractional solution $x_1 = x_2 = \ldots = x_m$ and $\sum_{j \in [m]} F(x_j) \geq (1 - e^{-1}) \max_{\pi \in \Pi(V, m, \mathcal{M})} \sum_{A \in \pi} f(A)$.

Since the fractional solution are guaranteed to be in the matroid polytope of $\mathcal{M}$ and $\mathcal{M}_m^p$, we have $\forall i \in [n], j \in \{1, \ldots, m\}, x_j[i] \leq \frac{1}{m}$ and $\sum_{i \in V} x_j[i] \leq r_{\mathcal{M}}(V)$.

$\square$

**Lemma 7 (Matroid Constraint Round-robin with No Large Singletons).** *Suppose for all $v \in V$, we have $f(v) \leq \frac{1-e^{-1}}{5} OPT^{\mathcal{M}}$, where $OPT^{\mathcal{M}}$ is the optimal solution value of the robust submodular partition problem constrained by matroid $\mathcal{M}$ (in other words, all the singletons have relatively small values for the given problem instance). Then, the round-robin iterations of Alg. 7 (line 10) gives a solution $\min_{j \in [m]} f(A_j) \geq \frac{1-e^{-1}}{5} OPT^{\mathcal{M}}$.*

---

**Algorithm 7:** Matroid Round-Robin Greedy

**input** : $f, V, m$, matroid constraint $\mathcal{M}$, discounting factor for guessing optimal $\delta$

1 Let $\tau$ be the solution value of Alg. 3;
2 Let $high = \lceil \log_{1+\delta}(m + 2) \rceil$, $low = 0$;
3 Create a sequence of guessed values: $(\tau, (1 + \delta)\tau, (1 + \delta)^2\tau, \ldots, (1 + \delta)^{high}\tau)$;
4 Create an empty solution ($\emptyset$ for each block in the partition) for each guessed value $\pi_0, \pi_1, \ldots, \pi_{high}$;
5 **while** $high \geq low$ **do**
6      Let $idx = \lfloor (high + low)/2 \rfloor$; Let $A_1 = A_2 = \ldots = A_m = \emptyset$;
7      Let $V' = \{v | v \in V, f(v) \leq \frac{1-e^{-1}}{5}(1 + \delta)^{idx}\tau\}$; Let $G = V \setminus V'$;
8      Assign $G$ to $A_{m-|G|+1}, A_{m-|G|+2}, \ldots, A_m$ with one element per block;
9      Let $m' = m - |G|$;
10      Let $\{A_1, A_2, \ldots, A_{m'}\} = RR(f, V', m', \mathcal{M}, [m'])$;
11      **if** $f(A_j) \geq \frac{(1-e^{-1})}{5}(1 + \delta)^{idx}\tau \ \forall j \in [m']$ **then**
12          Let $\pi_{idx} = \{A_1, A_2, \ldots, A_m\}$; Let $low = idx + 1$;
13      **else**
14          Let $high = idx - 1$;
15 **return** best of $\pi_0, \pi_1, \ldots, \pi_{high}$;

---

*Proof.* Let's focus on one block (any one in $A_1, \ldots, A_{m'}$) and for simplicity, we will omit the block index $j$ for this proof if not further noticed. Denote $OPT = \min_{j \in [m]} f(O_j^{\mathcal{M}})$ for this proof. Also, we assume we know the optimal solution value $OPT$ for this proof. Note that in the complete version of Alg. 7, we need to remove large singleton values based on the guessed optimal value, but for this lemma we make the assumption that in the given problem instance, there are no large singletons present.

For the current block, we denote the final resulting set from Alg. 7 as $A$. For one round-robin iteration, we go over all the feasible blocks sequentially, and to get $A$ we need to run $|A| = r$ round-robin iterations. Note that for different blocks, the number of round-robin iterations might be different.

We then divide the restricted ground set $V'$ by the round-robin iterations with respect to the current block $A$. Before we add the first element to $A$, denote all the allocated elements by $V^0$. Then we can think that for every round-robin iteration, we always start from the current block $A$. Let $V' = V^0 \cup V^1 \cup \ldots \cup V^r$ be a partition of $V'$ and $V^t$ contains all the elements allocated during the $t$'s round-robin iteration. Note $V^r$ contains all the unallocated elements in the ground set after we add the last element to $A$. Let $V^{t_1:t_2} = \cup_{t \in \{t_1, t_1+1, \ldots, t_2\}} V^t$. Accordingly, we partition the result $A$ by $A^t = A \cap V^t$.

For the set $V' \setminus A$, we separate it into two parts $Q'_1$ and $Q'_2$, where $Q'_1$ contain all the elements checked in Alg. 7 that cannot be added to the current block due to the matroid constraint, and let $Q'_2 = V' \setminus A \setminus Q'_1$, i.e., $Q'_2$ contain all the elements that can be added the current block. To be more precise:

$$Q'_1 = \cup_{t \in \{0,1,\ldots,r\}} \cup_{v \in V^t \setminus A^t, (A^{1:t} \cup v) \notin \mathcal{M}} v \tag{40}$$

Let $Q_1 = Q'_1 \cup A$ and $Q_2 = Q'_2 \cup A$.

Let $F$ denote the multilinear extension of $f$, i.e., $F(x) = \mathbb{E}_{R \sim x} f(R)$. By Lemma 5 and Lemma 6, we know that

$$(1 - e^{-1})OPT \leq \max_{\pi \in \Pi(V', m', \mathcal{M})} \min_{S \in \pi} f(S) \tag{41}$$

$$\leq \frac{1}{m'} \max_{\pi \in \Pi(V', m', \mathcal{M})} \sum_{S \in \pi} f(S) \tag{42}$$

$$\leq F(x) \tag{43}$$

$$\leq (F(x \cap Q_1) + F(x \cap Q_2)). \tag{44}$$

Note that $x$ is the fractional solution to the continuous greedy algorithm on the welfare objective: $\max_{\pi \in \Pi(V', m', \mathcal{M})} \sum_{S \in \pi} f(S)$ (similar to one of the $x_j$'s in Lemma 6 and we omit the block index for this proof). Here we use $x \cap Q$ to represent setting all elements not in $Q$ to be zero in the $x$ fractional solution. The first inequality follows from Lemma. 5. The second inequality follows that the sum over blocks of the max-min solution is no better than the optimal solution of the welfare problem. Since every element is sampled independently according to its probability in the fractional solution $x$, together with submodularity ($Q_1 \cup Q_2 = V'$) we get the last inequality above. Next, we will bound the two terms $F(x \cap Q_1)$ and $F(x \cap Q_2)$ separately.

For the first term $F(x \cap Q_1)$, we know that $r = r_{\mathcal{M}}(Q_1)$, and Alg. 7 generates $A$ in the same manor as running greedy max on $Q_1$ with matroid constraint $\mathcal{M}$. To be more precise, suppose $\mathcal{M} = (V, \mathcal{I})$ and we remove all the elements that are not in $Q_1$ and get $\mathcal{M}' = (V' \cap Q_1, \{I \cap Q_1 \forall I \in \mathcal{I}\})$. Note that $\mathcal{M}'$ is also a matroid, and all sets that satisfy $\mathcal{M}'$ also satisfy $\mathcal{M}$ due to the down-monotone property of matroids. Therefore, we have:

$$f(A) \geq \frac{1}{2} \max_{S \in \mathcal{M}'} f(S) \tag{45}$$

$$\geq \frac{1}{2} F(x \cap Q_1). \tag{46}$$

Note that $x$ is in the matroid polytope of $\mathcal{M}$, and $x \cap Q_1$ is in the matroid polytope of $\mathcal{M}'$. By pipage rounding, we know that we can get an integral solution $X'$ from $F(x \cap Q_1)$ so that the integral solution still satisfies $X' \in \mathcal{M}'$ and $f(X') \geq F(x \cap Q_1)$. Since $\max_{S \in \mathcal{M}'} f(S) \geq f(X')$, we get the last inequality above.

For the second term $F(x \cap Q_2)$, we will bound it using the greedy step. Denote $y = x \cap Q_2$, $y^t = y \cap V^t$, and $\mathbb{E}_S(y) = \mathbb{E}_{R \sim y} f(R|S)$, we have:

$$F(y) = \mathbb{E}_{R \sim y} f(R) \tag{47}$$

$$= \mathbb{E}_{R \sim y^0} f(R) + \mathbb{E}_{R^1 \sim y^0, R^2 \sim y^{1:r}} f(R^2|R^1) \tag{48}$$

$$\leq F(y^0) + F(y^{1:r}) \tag{49}$$

$$\leq F(y^0) + f(A^1) + \mathbb{E}_{A^1}(y^{1:r}) \tag{50}$$

$$= F(y^0) + f(A^1) + \mathbb{E}_{A^1}(y^1) + \mathbb{E}_{R^1 \sim y^1, R^2 \sim y^{2:r}} f(R^2|R^1 \cup A^1) \tag{51}$$

$$\leq F(y^0) + f(A^1) + \mathbb{E}_{A^1}(y^1) + \mathbb{E}_{A^1}(y^{2:r}) \tag{52}$$

$$\leq F(y^0) + f(A^1) + \mathbb{E}_{A^1}(y^1) + f(A_2|A_1) + \mathbb{E}_{A^2}(y^{2:r}) \tag{53}$$

Continue to unwrap $\mathbb{E}_{A^2}(y^{2:r})$ in the same way, finally we get:

$$F(y) \leq F(y^0) + \left[ f(A^1) + f(A^2|A^1) + f(A^3|A^2) + \ldots + f(A^r|A^{r-1}) \right]$$
$$+ \left[ \mathbb{E}_{A^1}(y^1) + \mathbb{E}_{A^2}(y^2) + \ldots + \mathbb{E}_{A^r}(y^r) \right] \tag{54}$$

$$= F(y^0) + f(A) + \left[ \mathbb{E}_{A^1}(y^1) + \mathbb{E}_{A^2}(y^2) + \ldots + \mathbb{E}_{A^r}(y^r) \right] \tag{55}$$

We then need to bound $F(y^0)$ and $\left[ \mathbb{E}_{A^1}(y^1) + \mathbb{E}_{A^2}(y^2) + \ldots + \mathbb{E}_{A^r}(y^r) \right]$. Note that by assumption, we do not have any singleton gains larger than $\frac{1-e^{-1}}{5}$, and we have:

$$F(y^0) \leq \frac{1-e^{-1}}{5} OPT \tag{56}$$

Since we select items greedily at every round-robin step, and $y$ only has non-zero values for elements that are in $Q_2$, we have:

$$\mathbb{E}_{A^t}(y^{t+1}) = \mathbb{E}_{R \sim y^{t+1}} f(R|A^t) \tag{57}$$

$$\leq \sum_{v \in y^{t+1}} y^{t+1}(v) f(v|A^t) \tag{58}$$

$$\leq \sum_{v \in y^{t+1}} \frac{1}{m'} f(v|A^t) \tag{59}$$

$$\leq f(A^t|A^{t-1}) \tag{60}$$

Note that for the last round-robin iteration $V^r$, it may seem that there can be more than $m'$ elements, but it's not possible: since there are no new elements added to the current blocks, $V^r \cap Q_2$ contains at most $m'$ elements as otherwise we will find new feasible elements and add to block $A$.

Then we sum over all $t$ and get:

$$\left[ \mathbb{E}_{A^1}(y^1) + \mathbb{E}_{A^2}(y^2) + \ldots + \mathbb{E}_{A^r}(y^r) \right] \leq f(A). \tag{61}$$

Therefore, we have:

$$(1-e^{-1})OPT \leq (F(x \cap Q_1) + F(x \cap Q_2)) \tag{62}$$

$$\leq 2f(A) + F(y) \tag{63}$$

$$\leq 2f(A) + 2f(A) + \frac{1-e^{-1}}{5} OPT \tag{64}$$

$$f(A) \geq \frac{1-e^{-1}}{5} OPT. \tag{65}$$

$\square$

Next, we will discuss why binary search can be used in the guessing of the optimal value (as opposed to the case of linear search where we try all possible guessed optimal values). We make almost the same argument as the cardinality constraint case. We restate the argument and proof here for completeness.

**Lemma 8 (Binary Search).** *For Alg. 7, let the potential guessed optimal values form a sequence $(\tau_1, \tau_2, \ldots, \tau_l)$ where $\tau_{i+1} = (1 + \delta)\tau_i$. Then for any $\tau_i \leq OPT^{\mathcal{M}}$ as the guessed optimal value that we plug into the binary search iterations (Alg. 7 line 7-14), line 14 of Alg. 7 is always true.*

*Proof.* For simplicity, denote the optimal solution of the matroid constrained robust partitioning problem as $OPT$ for this proof. Let's denote the found large singleton values and the remaining sets (line 7 of Alg. 7) respectively by $G_{OPT}$ and $V'_{OPT}$ for $OPT$, and $G_i$ and $V'_i$ for $\tau_i$. Since $\tau_i \leq OPT$, $G_{OPT} \subseteq G_i$ as the threshold is smaller for $\tau_i$. Then by Lemma. 5, we know that the optimal solution $OPT_i$ on $V'_i$ (partitioned to $m - |G_i|$ blocks) is no less than the optimal solution $OPT'$ on $V'_{OPT}$ (partitioned to $m - |G_{OPT}|$ blocks). Also note that since we remove all singleton values to form $V'_i$ based on the threshold $\frac{1-e^{-1}}{5}\tau_i$, and $\tau_i \leq OPT \leq OPT' \leq OPT_i$ ($OPT \leq OPT'$ is also from Lemma. 5), we are guaranteed that there are no singleton elements with $f(v) \geq \frac{1-e^{-1}}{5}OPT_i$ in $V'_i$. Therefore, based on Lemma. 7, our round robin iterations on $V'_i$ give a solution whose every block has a value of at least $\frac{1-e^{-1}}{5}OPT_i \geq \frac{1-e^{-1}}{5}\tau_i$, and thus line 11 is guaranteed to be true for $\tau_i$. $\square$

Based on Lemma. 8, either of the following must hold: 1) for all $\tau_i > OPT^{\mathcal{M}}$ line 11 is false, and in such a case, we can use binary search to find the largest $\tau_i$ with line 11 true, and let's call it $\tau_{i^*}$; 2) there exists some $\tau_i > OPT^{\mathcal{M}}$ such that line 11 is true. If we find such $\tau_i$, we find a solution with $f(A_j) \geq \frac{1-e^{-1}}{5}\tau_i \geq \frac{1-e^{-1}}{5}OPT^{\mathcal{M}}$. Otherwise if we don't find it, we go to the first case and will still find $\tau_{i^*}$.

Finally, the approximation guarantee of Alg.7 follows by combining the previous lemmas.

**Theorem 2 (Matroid Constrained Round-Robin).** *For the problem in Eq. (4), with $\mathcal{C}$ as any matroid constraint $\mathcal{M}$, Alg 7 gives a solution $\min_{j \in [m]} f(A_j) \geq \frac{(1-e^{-1})}{5} \min_{j \in [m]} f(O_j^{\mathcal{M}})$. The time complexity of of Alg. 7 is $\mathcal{O}(n^2(\log \log m + \log \frac{1}{\delta}))$.*

*Proof.* Firstly, based on the previous arguments about the binary search, we can find a $\tau_i$ with $\tau_i \geq \tau_{i^*}$, where $\tau_{i^*}$ is the largest $\tau_j$ with $\tau_j \leq OPT^{\mathcal{M}}$ and line 11 of Alg. 7 true. By setting $\delta$ small, $\tau_{i^*}$ can be arbitrarily close to $OPT^{\mathcal{M}}$. Next, based on Lemma. 7, after removing the large singleton values, the solution on the remaining elements have the min block value at least $\frac{1-e^{-1}}{5}\tau_i$, and the removed large singletons are all larger than $\frac{1-e^{-1}}{5}\tau_i$ by construction. We therefore get the approximation ratio. $\square$

## C Synthetic Experiment

We compare Alg. 3, Alg. 7 (10 optimal value guesses) and the random assignment baseline on randomly generated synthetic facility location functions. Every entry in the facility location similarity matrix is uniformly sampled from $[0, 1]$. We report our results with different parameters in Figure 4.

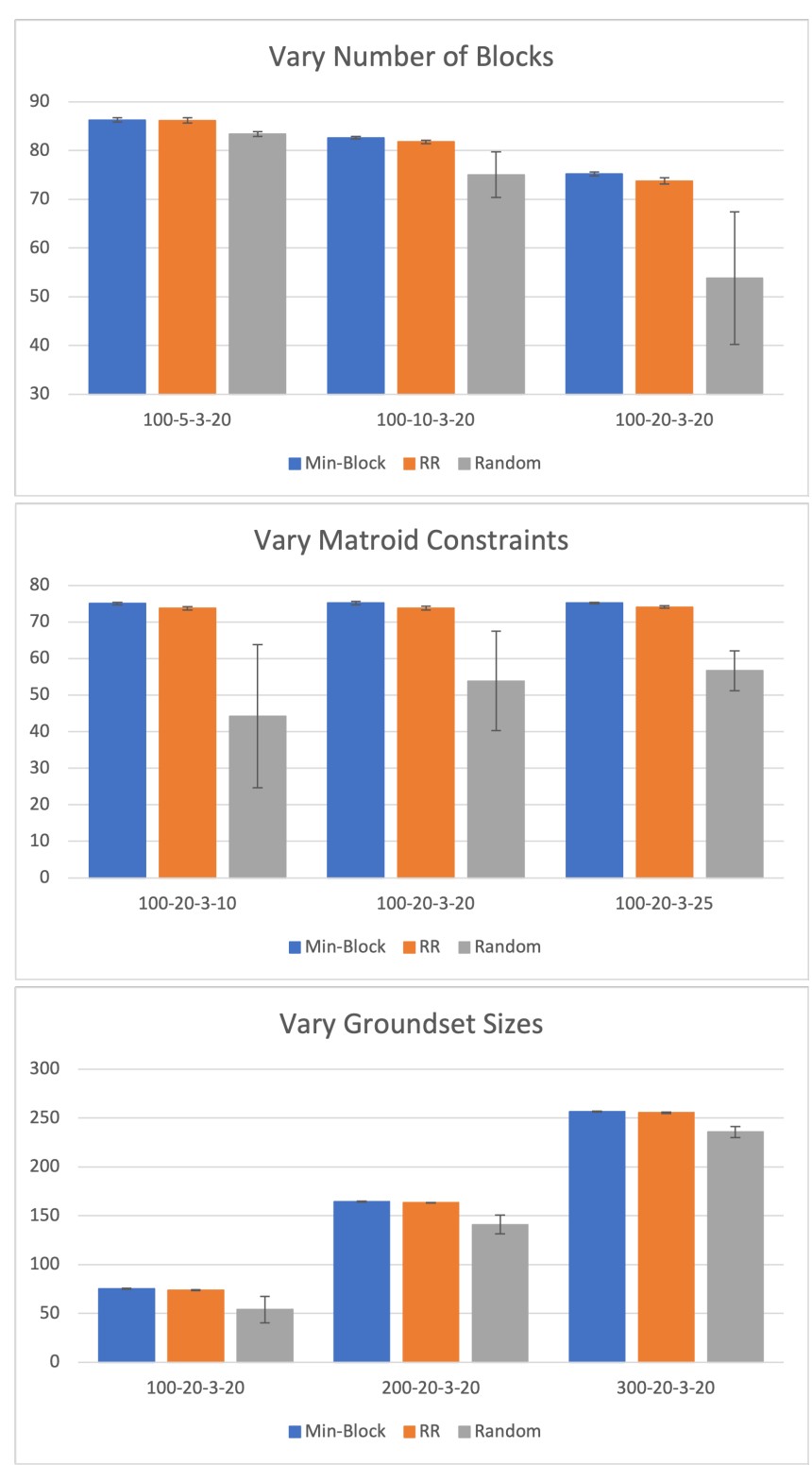

**Figure 4:** Synthetic data results on randomly generated facility location function similarity matrices. The $x$−label is $n − m − c − p$. Results averaged over 30 runs.

Particularly, we have four parameters for every problem instance: $n$ the ground set size, $m$ the number of partitions, and the partition matroid constraint parameters $c$ and $p$. For the partition

matroid constraint, we divide the groundset into blocks of size $p$ and for every such block, it is constrained that we can pick at most $c$ elements. For all variants of the settings, we observe that Alg. 3 and Alg. 7 significantly outperform the random baseline. Alg. 3 consistently outperforms Alg. 7 but the margin is not very large.

## D  Experiment Details

The features used in the experiments are generated through an autoencoder. The network architecture is described in Table 1. The network is trained using ReLU non-linearity and batch normalization. ADAM [15] is utilized as the optimization method with an initial learning rate of 5e-3, a weight decay of 5e-4 and a minibatch size of 100. The network is trained in PyTorch using the procedure described in [20]. Features are extracted as the output of the autoencoder's bottleneck (following the residual block and non-linearity).

The training of the ResNet-9 (Myrtle-AI, `https://github.com/davidcpage/cifar10-fast`) network utilizes an ADAM optimizer with an initial learning rate of $1e-3$. The network is trained for 90 epochs. For CPU jobs, we use a single core Intel Xeon 2.10GHz CPU, and for GPU jobs we use a Nvidia RTX 2080Ti GPU.

**Table 1:** Neural network structure of the autoencoder

| Group | Block Type (kernel sz, stride, channels) | # Blocks |
|---|---|---|
| conv1 | [ $3 \times 3$ ], 2, 64 | 1 |
| conv1 (residual) | $\begin{bmatrix} 3 \times 3 \\ 3 \times 3 \end{bmatrix}$, 1, 64 | 2 |
| conv2 | [ $3 \times 3$ ], 2, 16 | 1 |
| conv2 (residual) | $\begin{bmatrix} 3 \times 3 \\ 3 \times 3 \end{bmatrix}$, 1, 16 | 2 |
| conv3 | [ $3 \times 3$ ], 2, 8 | 1 |
| conv3 (residual) | $\begin{bmatrix} 3 \times 3 \\ 3 \times 3 \end{bmatrix}$, 1, 8 | 2 |
| conv4 | [ $3 \times 3$ ], 1, 4 | 1 |
| conv4 (residual) | $\begin{bmatrix} 3 \times 3 \\ 3 \times 3 \end{bmatrix}$, 1, 4 | 1 |
| deconv4 (residual) | $\begin{bmatrix} 3 \times 3 \\ 3 \times 3 \end{bmatrix}$, 1, 4 | 1 |
| deconv3 | [ $3 \times 3$ ], 1, 8 | 1 |
| deconv3 (residual) | $\begin{bmatrix} 3 \times 3 \\ 3 \times 3 \end{bmatrix}$, 1, 8 | 2 |
| deconv2 | [ $3 \times 3$ ], 2, 16 | 1 |
| deconv2 (residual) | $\begin{bmatrix} 3 \times 3 \\ 3 \times 3 \end{bmatrix}$, 2, 16 | 2 |
| deconv1 | [ $3 \times 3$ ], 2, 64 | 1 |
| deconv1 (residual) | $\begin{bmatrix} 3 \times 3 \\ 3 \times 3 \end{bmatrix}$, 2, 64 | 2 |
| deconv0 | [ $3 \times 3$ ], 2, 3 | 1 |

We also test the case for various cardinality constraints on CIFAR-10 datatset, and report the results in Figure 5. The running times (in seconds) for Figure 5 and Figure 1 is given in Table 2 and Table 3 respectively under various constraint parameters. For the round-robin algorithm, we pick $\delta = 0.1$. The computing platform uses a single core of Intel Xeon(R) CPU 2.10GHz.

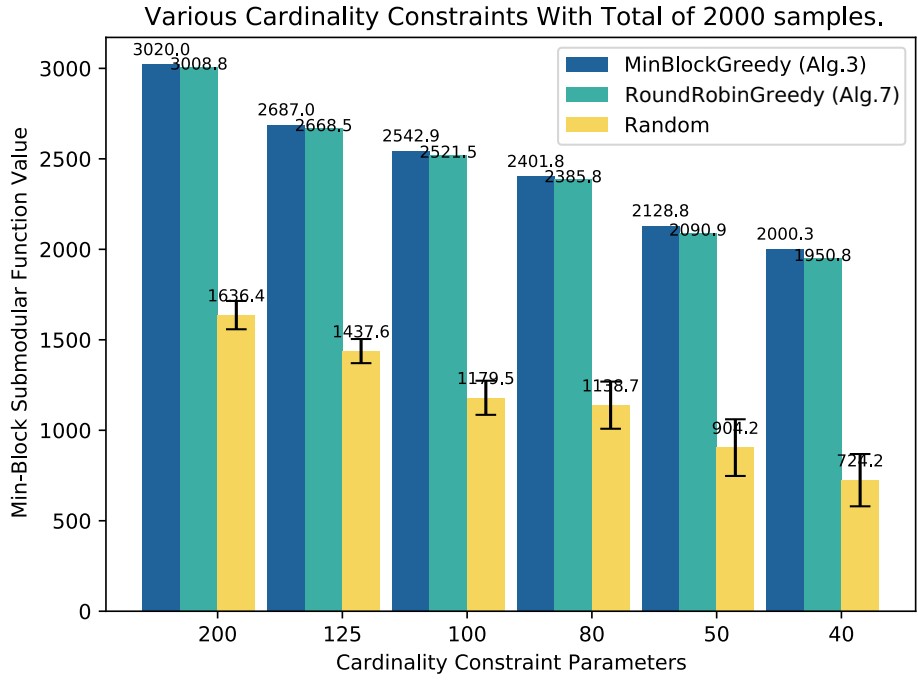

**Figure 5:** Cardinality constrained results. We select a total of 2000 samples and the number of blocks varies according to the constraint, e.g., when the constraint $k = 200$, $m = 2000/200 = 10$.

**Table 2:** Wall Clock Running Time for Figure 5

| Constraint Param | 200 | 125 | 100 | 80 | 50 | 40 |
|---|---|---|---|---|---|---|
| Min-Block Greedy | 60.7 | 60.4 | 60.0 | 57.9 | 57.0 | 57.1 |
| Round-Robin Greedy | 240.2 | 233.5 | 243.7 | 244.6 | 240.0 | 242.1 |

**Table 3:** Wall Clock Running Time for Figure 1

| Constraint Param | 40 | 25 | 20 | 10 | 8 | 5 |
|---|---|---|---|---|---|---|
| Min-Block Greedy | 63.4 | 62.3 | 62.2 | 60.0 | 59.5 | 59.9 |
| Round-Robin Greedy | 184.2 | 180.3 | 182.1 | 242.4 | 244.0 | 240.0 |