# OpenReview forum: "Constrained Robust Submodular Partitioning"
_NeurIPS.cc/2021/Conference — NeurIPS 2021 Spotlight_

### Official Review · Reviewer_cEdN · 2021-07-05

**Rating:** 7
**Confidence:** 4

**Summary:**

Provides constant approximation algorithms for certain "robust" or min-max submodular maximization problems. They consider cardinality and matroid constraints. Also provide good experimental results on partitioning a training dataset.

**Limitations And Societal Impact:**

yes

**Main Review:**

The paper is on a “robust” submodular maximization problem. There is a monotone submodular function f and matroid C on items. Allocate items to m blocks so that the items in each block are independent in C, and maximize the minimum f(S) over all blocks.

The results in the paper are-
1)	If C is a cardinality constraint, there is a (1-1/e)^2/3 approximation.
2)	If C is any matroid, there is a (1-1/e)/5 approximation.
3)	If C is more general (with any A-approximation for submod max under C) then there is ~ A/m approximation.
4)	Experiments on a real-world training data set with classes, that corresponds to C = partition matroid. The proposed algorithms achieve much better objective than a random solution. They also train the partitioned data and report the test accuracy- solutions with high objective have better accuracy.

I found results 1 & 2 theoretically interesting. It relies heavily on ideas for the unconstrained problem from [3] where a (1-1/e)/3 approximation was obtained. However, there are some non-trivial technical issues to deal with for the matroid constraint. The authors also emphasize that they get a strongly polynomial runtime- this just seems to use standard techniques.

I do not find result 3 (in sec 2) interesting. The theoretical bound is weak (linear in #blocks), although it applies to any constraint.

Why don’t you focus on results 1-2 and provide more details in the main body? Also, does the matroid analysis extend to p-matroid intersection?
--
After the response, I still think this is a good paper.


**Time Spent Reviewing:**

6

---

> ### Author Response · Authors · 2021-08-10
> **Response to Reviewer cEdN**
>
> Again, thanks for your time and suggestions!
>
> About focus on results 1 & 2: we will add more discussions about the proofs for results 1 & 2 in the next version.
>
> About a generalization to p-matroids: unfortunately, we don't think the current approach can be easily extended to the p-matroid case, as it's hard to argue about the number of items that each block obtains. As opposed to the case of a single matroid, we know that the number is around the rank of the matroid. To extend it to a p-matroid, we probably need to use matroid exchange techniques. This also shows the value of result 3, as currently it's the only algorithm that can work for the more general constraint cases.

---

### Official Review · Reviewer_SBUL · 2021-07-13

**Rating:** 8
**Confidence:** 4

**Summary:**

The focus of this work is constraint robust submodular maximization. In this problem, the goal is to find $m$ sets maximizing the minimum value of a given submodular function $f$ over these sets. This problem has attracted attention recently and  algorithms are designed for offline and streaming setting. In this work, author consider this problem subject to three famous constraint and provide two type of algorithms.

1- Min-block greedy algorithms: This algorithm has been used before for non-constraint version robust submodular maximization and provides a $1/m$ approximate solution.  In this work they extend it to the constrained version and achieve $\frac{\alpha}{\alpha m+1}$ approximation algorithm, where $\alpha$ is the approximation guarantee of the greedy offline algorithm for that constraint. Intuitively this result is very close to being tight with respect to the unconstrained for large $m$ values.

2- Round Robin greedy algorithm: They present a $\frac{(1-e^{-1})^2}{3}$-approximation algorithm for the cardinality constraint and  $\frac{1-e^{-1}}{5}$ for the matroid constraint.

**Limitations And Societal Impact:**

yes.


**Main Review:**

I have mostly positive comments for this work. The results are interesting and ideas used are novel. At the first glance it might look like that the contribution is marginal given that their approach is similar to the unconstrained algorithm. Extending those ideas to constraint setting is non-trivial and requires work. The paper is in great shape, it is easy to follow and the similarity with the previous work is clearly stated.  The algorithms are explained step by step which makes it even more understandable. The experimental setting is also clearly stated and the results are promising.  The authors have only compared their approach against random solution, which is expected as to the best of my knowledge there are no such algorithms known before. I enjoyed Lemma 2, when reading the paper it came to my mind and I was happy to see that the authors have considered it. Assuming that I am not missing any previous work on this paper, I recommend this paper to be accepted.

A few minor comments:
- I recommend the authors to extend the theorems  to mention the number of oracle calls as well. It gives another reason why the algorithm in section 4 is interesting when compared to section 5.

- The plots are hard to read, enlarging the font improves readability.

**Time Spent Reviewing:**

6 hours

---

> ### Author Response · Authors · 2021-08-10
> **Response to Reviewer SBUL**
>
> Thanks also for your time and suggestions!
>
> For comment one about adding the oracle calls/time complexity to the theorems: we will add them in the next version of the paper.
>
> For comment two about improving the presentation of the plots: we will make the plots and the fonts larger and easier to read in the next version.

---

> > ### Comment · Reviewer_SBUL · 2021-09-10
> > **Rebuttal**
> >
> > I acknowledge that I have read the rebuttal and other comments. I think the paper is nice and strong.

---

### Official Review · Reviewer_VawE · 2021-07-15

**Rating:** 7
**Confidence:** 3

**Summary:**

This paper studies constrained robust submodular partitioning. Given a submodular function $f$, a constraint set family $\\mathcal{C}$, and the number $m$ of blocks, we are to find an $m$-partition $(X_1, \\dots, X_m)$ of the ground set that maximizes $\\min_{i=1}^m f(X_i)$ subject to $X_i \\in \\mathcal{C}$.

The main contributions of this paper are two algorithms called Min-Block Greedy and Round-Robin Greedy:
- Min-Block Greedy achieves $\\alpha/(\\alpha m + 1)$-approximation for a general down-closed $\\mathcal{C}$, given an $\\alpha$-approximation algorithm for submodular function maximization on $\\mathcal{C}$.  Furthermore, the authors proved that the analysis is tight for the unconstrained case.
- Round-Robin Greedy achieves $(1-1/e)^2/3$-approximation when $\\mathcal{C}$ is a cardinality constraint. It also achieves $(1-1/e)/5$-approximation when $\\mathcal{C}$ is a matroid.

**Main Review:**

I enjoyed reading this paper. It is surprising (to me) that this max-min type problem under a matroid constraint still admits a constant-factor approximation. Furthermore, very simple algorithms surprisingly work! I think the main results are significant.

On the other hand, the writing is not so clear in several places. I have a technical question for Algorithm 5; see below. I found several typos around the description of Round-Robin Greedy, so possibly my questions are just due to the typo.
Another minor complaint is: there was only a sketch of the proof rather than a formal proof for Round-Robin Greedy with the cardinality constraint. The authors should fix these issues in a future version.

Overall, I believe the paper has significant contributions to submodular function maximization. But I would like to hear the authors' response before giving a high score.

## Technical comment
Algorithm 5, line 10: Should $\\max_{\\pi \\in \\Pi(V', m', k)}$ be $\\max_{\\pi \\in \\Pi(V', m')}$? Swap rounding only applies to a single matroid constraint. However, $\\Pi(V', m', k)$ is not a single matroid but the intersection of two matroids.

## Minor comments
- Algorithm 6, line 4: It seems that you need to require $v \\in R$.
- Algorithm 7, line 9: $V''$ is not defined. Is it the same as Algorithm 5?
- Algorithm 7, line 10: Should $RR(f, V'', m', \\mathcal{M}, [m'])$ be $RR(f, R, m', \\mathcal{M}, [m'])$?
- Appendix B: Proof of Round-Robin Greedy for the cardinality case is a sketch rather than a proof. You should formally prove a version of Lemma 3 of [3] involving $\\gamma$.
- Appendix B, eq (55): Why do you have $F(y^0) \\leq \\frac{1-1/e}{5}OPT$ when every singleton gain is small?

**Time Spent Reviewing:**

5

---

> ### Author Response · Authors · 2021-08-10
> **Response to Reviewer VawE**
>
> Thanks for your time and suggestions as well!
>
> For the technical question: please check Theorem III.3 in [7], where they show a bound for a general setting of a constrained allocation problem that can be achieved using a randomized swap rounding based approach. The general setting in [7] covers the case where we have an additional cardinality constraint for each block in the allocation. We will make this more clear in the next version of the paper.
>
>
> Minor comments:
>
> Alg 6 line 4: yes, we will add it.
>
> Alg 7 line 9: this is a typo, and it should be $V'$. Thanks for catching this.
>
> Alg 7 line 10: yes, you are right.
>
> More formal proof for Lemma 3: yes, we will add a more detailed and formal proof in the next version.
>
> Appendix B eq(55): that is because $F(y^0)$ is upper bounded by the largest singleton value. In more detail, $y^0$ is a continuous vector with all entries summed to one, and due to submodularity and the definition of the multi-linear extension $F$ (which can be seen as an expected value of f with items sampled from $y^0$), its value cannot exceed $\max_{v \in V} f(v)$.

---

### Official Review · Reviewer_JRJn · 2021-07-16

**Rating:** 7
**Confidence:** 5

**Summary:**

This paper focuses on the problem of finding an allocation of $n$ items in $m$ blocks that maximizes the minimum of the block valuations, where each block has the same monotone submodular objective (homogeneous case) and the subset allocated to each block has to satisfy a feasibility constraint. The model allows allocations that do not necessarily assign every item, since each block allocation needs to be feasible.

The authors first studied the Min-Block Greedy (introduced in [29]) which in each iteration selects the block of minimum value and greedily assigns a new element. They show that the $(1/m)$-guarantee (proved in [29]) of this algorithm is tight. Then, they extended this algorithm to the constrained version and prove a $\alpha/(m\alpha+1)$ factor, where $\alpha$ is the factor for the vanilla constrained submodular maximization problem. In the second part, the authors give a Round-Robin Greedy algorithm (which is adapted from the algorithm in [3]) that achieves a $(1-1/e)^2/3$ factor for the cardinality constrained case. This algorithm is later extended to the matroid case for which it achieves a $(1-1/e)/5$ guarantee. Finally, they present computational experiments to compare the proposed algorithms.


**Limitations And Societal Impact:**

the authors address the limitations and potential negative societal impacts

**Main Review:**

To the best of my knowledge, this work is the first to provide constant factors for the model with more general constraints. The unconstrained homogenous case was previously studied in [3] in which they obtained a (1-1/e)/3 factor. This work adapted the algorithm in [3] to obtain constant guarantees for the constrained version of the problem. The paper is overall well written. The proofs are correct, I carefully checked most of them. In particular, the proof of Lemma 3 follows from the result in [3] plus the fact that Algorithm 5 uses the continuous greedy. The proof of Theorem 2 seems to be an adaptation of the proof in [3]. The authors should emphasize the differences with [3] or the challenges in the analysis for matroid constraints.

Some comments/questions:
-	The authors should comment that [3] and [11] focus on a different type of guarantee. Specifically, they consider the maximin share as a benchmark which differs from the one considered in this paper in the heterogenous case, but it coincides in the homogenous case. This is crucial since certain parts of the proofs rely on the results obtained in [3], for example, (32) in the Appendix. Part of the discussion in Section C in the Appendix could be in the main body as well.
-	In Section 6 (page 9 line 373), the authors say that the performance of Algorithm 7 is affected by the ordering of the blocks. Is there a way to heuristically optimize this or via preprocessing?

Minor comments:
-	Page 3 line 120: properties of matroid are missing.
-	Page 4 line 157: Where -> where
-	Page 5 eq (5):  period instead of comma.
-	Page 6 Corollary 2: this should be appropriately cited since it was already proved in [25]
-	Page 7 line 288: $f(O_j)$
-	Page 7 Algorithm 5: line 11 $A_{m’}$
-	Page 8 Algorithm 6: lines 8 and 9 outside the if? Also, line 10 should be $A_{|J|}$ which is not necessarily $A_m$.
-	Page 8 Algorithm 7: $R=V’’$ not used, in the input of RR is V’ instead of V’’. Line 10 should be $A_{m’}$.
-	Page 9 line 361: is it a cardinality constraint?
-	Page 9 Figure 1: legend should say Alg 7?
-	Appendix line 660: We assume. Same in line 749
-	Appendix lines 702-721: it would be good to comment on this in the main body
-	Appendix: Algorithms are renumbered.
-	Appendix line 853: $\{1,\ldots,m\}$


**Time Spent Reviewing:**

15

---

> ### Author Response · Authors · 2021-08-10
> **Response to Reviewer JRJn**
>
> Thanks for your time and suggestions!
>
> For the first question about different objectives compared to [3] and [11]: we will add the discussion about the homogeneous/heterogeneous case and the objective in [3] and [11] as suggested. We will add discussion in Appendix C to the main paper as well.
>
> For the second question about the different ordering of the blocks: a possible way to make the blocks more balanced is to change the round-robin ordering to a back-and-forth manner, i.e., we go from the first block to the last block (in some arbitrary order), then go from the last block to the first and repeat. This approach may also exhibit some other theoretical guarantee, but it's probably weaker than the original algorithm's bound. Another possible trick can be to have a random ordering of the blocks for each round-robin iteration. This may also have some theoretical guarantee in expectation but also tends to be worse than the proposed one.
>
> Thanks for catching the typos in minor comments! We will correct them in the next version. For "Page 9 line 361": it is a partition matroid constraint as it limits the number of samples for each class of the CIFAR dataset. For "Page 9 Figure 1": yes, you are right, and it should be Alg 7.

---

### Decision · Program_Chairs · 2021-09-27

**Decision:**

Accept (Spotlight)

**Comment:**

The reviewers find that the theoretical results in this paper are strong and that there are interesting novel ideas. In particular, the reviewers agree that adding constraints to the robust submodular partitioning problem introduces significant technical challenges. Overall, this paper makes significant progress to the area of robust submodular maximization.